# Fibroblast-derived *Hgf* controls recruitment and expansion of muscle during morphogenesis of the mammalian diaphragm

Elizabeth M Sefton[1]*, Mirialys Gallardo[1], Claire E Tobin[1], Brittany C Collins[1], Mary P Colasanto[1], Allyson J Merrell[2], Gabrielle Kardon[1]*

[1]Department of Human Genetics, University of Utah, Salt Lake City, United States; [2]Ambys Medicines, South San Francisco, United States

**Abstract** The diaphragm is a domed muscle between the thorax and abdomen essential for breathing in mammals. Diaphragm development requires the coordinated development of muscle, connective tissue, and nerve, which are derived from different embryonic sources. Defects in diaphragm development cause the common and often lethal birth defect, congenital diaphragmatic hernias (CDH). HGF/MET signaling is required for diaphragm muscularization, but the source of HGF and the specific functions of this pathway in muscle progenitors and effects on phrenic nerve have not been explicitly tested. Using conditional mutagenesis in mice and pharmacological inhibition of MET, we demonstrate that the pleuroperitoneal folds (PPFs), transient embryonic structures that give rise to the connective tissue in the diaphragm, are the source of HGF critical for diaphragm muscularization. PPF-derived HGF is directly required for recruitment of MET+ muscle progenitors to the diaphragm and indirectly (via its effect on muscle development) required for phrenic nerve primary branching. In addition, HGF is continuously required for maintenance and motility of the pool of progenitors to enable full muscularization. Localization of HGF at the diaphragm's leading edges directs dorsal and ventral expansion of muscle and regulates its overall size and shape. Surprisingly, large muscleless regions in *HGF* and *Met* mutants do not lead to hernias. While these regions are likely more susceptible to CDH, muscle loss is not sufficient to cause CDH.

*For correspondence:
sefton@genetics.utah.edu (EMS);
gkardon@genetics.utah.edu (GK)

**Competing interest:** The authors declare that no competing interests exist.

## Editor's evaluation

It was previously known that HGF and Met control delamination and migration of muscle progenitor cells that colonize the diaphragm. The article by Sefton and coworkers confirms and extends these observations using conditional mouse lines in which the HGF gene was targeted by Cre/loxP recombination, and Met inhibitors that are applied at different stages of development. Together these new data show that HGF derived from the pleuroperitoneal folds is directly required for the recruitment of Met+ muscle progenitors to the diaphragm and continues to be essential for the survival of a pool of progenitors that eventually form the diaphragm muscle. Moreover, the authors show the effects of the diaphragm muscle on the development of the phrenic nerve that innervates this muscle, in particular, in the absence of muscle deficits in the branching of the phrenic nerve are observed. Overall, the technical quality of the data on diaphragm muscle development and its effect on the branching of the phrenic nerve are excellent.

## Introduction

The diaphragm is an essential skeletal muscle and a defining feature of mammals (*Perry et al., 2010*). Contraction of the diaphragm, lying at the base of the thoracic cavity, powers the inspiration phase of respiration (*Campbell et al., 1970*). The diaphragm also serves an important passive function as a barrier separating the thoracic from the abdominal cavity (*Perry et al., 2010*). Respiration by the diaphragm is carried out by the domed costal muscle, composed of a radial array of myofibers surrounded by muscle connective tissue, extending laterally from the ribs and medially to a central tendon, and innervated by the phrenic nerve (*Merrell and Kardon, 2013*). Diaphragm development requires coordination of multiple embryonic tissues: (1) somites are well established as the source of the diaphragm's muscle (*Babiuk et al., 2003*; *Bladt et al., 1995*; *Dietrich et al., 1999*), (2) the cervical neural tube gives rise to the phrenic nerve (*Allan and Greer, 1997a*; *Allan and Greer, 1997b*), and (3) the pleuroperitoneal folds (PPFs), paired mesodermal structures located between the thoracic (pleural) and abdominal (peritoneal) cavities, form the muscle connective tissue and central tendon (*Merrell et al., 2015*). Integration of these three tissues into a functional diaphragm is critical, but how their development is coordinated and integrated is largely unknown.

Defects in the development of the diaphragm cause congenital diaphragmatic hernias (CDH), a common (1 in 3000 births) and costly ($250 million per year in the US) birth defect (*Pober, 2007*; *Raval et al., 2011*; *Torfs et al., 1992*). CDH compromises the integrity of the diaphragm by effecting muscularization, leading to an incomplete barrier between the abdominal and thoracic cavities. As a result, the liver herniates into the thorax, impeding lung development and resulting in long-term morbidity and up to 50% neonatal mortality (*Colvin et al., 2005*). Correct innervation of the diaphragm by the phrenic nerve is also essential, as breathing must be functional by birth and fetal breathing movements are important for normal lung development (*Jansen and Chernick, 1991*).

Recruitment of muscle progenitors and targeting of phrenic nerve axons to the nascent diaphragm are essential first steps for correct diaphragm development. The receptor tyrosine kinase signaling cascade initiated by the binding of the ligand hepatocyte growth factor (HGF) to its receptor MET is a promising candidate pathway for regulating these steps of diaphragm development as HGF/MET signaling has been implicated in multiple aspects of muscle and motor neuron development (reviewed by *Birchmeier et al., 2003*; *Maina and Klein, 1999*). HGF binding to MET leads to MET phosphorylation and the activation of multiple downstream pathways, including JNK, MAPK, PI3K/Akt, and FAK (*Organ and Tsao, 2011*). HGF/MET signaling is a critical regulator of muscle progenitors migrating from somites (*Bladt et al., 1995*; *Dietrich et al., 1999*; *Maina et al., 1996*). HGF is also critical for innervation. HGF acts as a chemoattractant, required for correct guidance of MET+ motor neuron axons to target muscles in the developing limb (*Ebens et al., 1996*; *Yamamoto et al., 1997*), and MET signaling is required for distinct functions in different motor neuron pools, including axon growth in the latissimus dorsi and motor neuron survival in the pectoralis minor (*Lamballe et al., 2011*). However, these reports do not distinguish between the effect of HGF on muscle versus nerve or rely on in vitro experiments. Thus, HGF/MET signaling has complex, tissue-specific roles in regulating the neuromuscular system.

Here we dissect the role of HGF/MET signaling in muscularization and innervation of the diaphragm. In previous studies, *Met* mutations have been associated with CDH (*Longoni et al., 2014*) and *Hgf* is downregulated in mutants or pharmacological treatments that induce diaphragmatic hernias in rodents (*Merrell et al., 2015*; *Takahashi et al., 2016*). Furthermore, *Met* null mice lack all diaphragm musculature (*Bladt et al., 1995*; *Dietrich et al., 1999*; *Maina et al., 1996*). However, which cells are the source of HGF and what steps of diaphragm muscle development HGF/MET signaling regulates is unclear. Using conditional mutagenesis, pharmacological treatments, and an in vitro primary cell culture system (*Bogenschutz et al., 2020*), we demonstrate that the diaphragm's connective tissue fibroblasts are a critical source of HGF that recruits and maintains MET+ muscle progenitors into and throughout the developing diaphragm. In addition, PPF-derived HGF, via its regulation of muscle, is required for defasciculation of the phrenic nerve. While either genetic or pharmacological inhibition of HGF/Met signaling results in large muscleless regions in the diaphragm, surprisingly these muscleless regions maintain their structural integrity and do not herniate. Thus, revising our previous conclusions (*Merrell et al., 2015*), we now show that muscle loss is not sufficient to induce CDH and additional connective tissue defects are required to weaken the diaphragm.

## Results

### *Hgf* and *Met* are expressed in the developing diaphragm

To begin dissecting the precise role(s) of HGF/MET signaling in diaphragm development, we investigated the expression of *Hgf* and *Met* in the early diaphragm. We first examined mouse embryos at embryonic day (E) 10.5 when muscle progenitors are migrating from cervical somites to the nascent diaphragm and progenitors have already populated the forelimb (*Sefton et al., 2018*). At this stage, *Hgf* is expressed in the mesoderm lateral to the somites (but not in the somites themselves) and in the limb bud mesoderm (*Figure 1A*), while *Met* is expressed in the somites and the muscle progenitors in the limb bud (*Figure 1D* and *Sonnenberg et al., 1993*). At E11.5, muscle progenitors have migrated into the PPFs of the diaphragm (*Sefton et al., 2018*). During this stage, *Hgf* is expressed throughout the pyramidal PPFs (*Figure 1B*, arrows), while *Met* is expressed in a more restricted region in the PPFs (presumably in muscle progenitors; *Figure 1E*, arrows, *Figure 1—figure supplement 1B*), in limb muscle progenitors (*Figure 1E* and *Sonnenberg et al., 1993*), as well as the phrenic nerve (*Figure 1F*). By E12.5, the PPFs have expanded ventrally and dorsally across the surface of the liver (*Merrell et al., 2015*; *Sefton et al., 2018*). Strikingly, *Hgf* is restricted to the ventral and dorsal leading edges of the PPFs (*Figure 1C*, arrows and asterisks). Although *Met* is no longer detectable by whole-mount RNA in situ hybridization, qPCR indicates that it is still expressed at E12.5 at comparable cycle threshold values to *Hgf* and *Pax7* (*Figure 1—figure supplement 1A*). These expression patterns suggest that HGF expressed in the mesoderm adjacent to the somites, PPF fibroblasts, and limb bud fibroblasts activates MET signaling in the diaphragm, limb muscle progenitors, and phrenic nerve.

### Fibroblast-derived *Hgf* and somite-derived *Met* are required for diaphragm and limb skeletal muscle

The complete absence of diaphragm and limb muscles in mice with null-mutations for *Met* (*Bladt et al., 1995*; *Dietrich et al., 1999*; *Maina et al., 1996*) demonstrates that *Met* is critical for the development of these muscles. The spatially restricted expression of *Met* and *Hgf* suggests that MET signaling in muscle progenitors is activated by HGF in the PPF and limb fibroblasts. Surprisingly, the tissue-specific requirement of *Met* and *Hgf* has not been genetically tested in vivo. We tested whether the receptor is required in somite-derived diaphragm and limb myogenic cells by conditionally deleting *Met* (*Huh et al., 2004*) via *Pax3^Cre^* mice (*Engleka et al., 2005*), which recombines in the somites, including all trunk myogenic cells. Consistent with a hypothesized critical role of MET in myogenic cells, conditional deletion of *Met* in the somitic lineage results in a muscleless diaphragm and limbs (*Figure 1I and J*, *Figure 1—figure supplement 1C and D*). Additionally, we tested whether HGF derived from PPF and limb fibroblasts is critical via *Pdgfra^CreERT2^* mice (*Chung et al., 2018*). *Pdgfra* is expressed in the PPFs of the diaphragm (*Figure 1—figure supplement 2B–D*), and *Pdgfra^CreERT2^* drives Cre expression in the connective tissue fibroblasts of the diaphragm and limb, but not in muscle fibers (*Figure 1—figure supplement 2E–L*). When combined with *Hgf^fl^* (*Phaneuf et al., 2004*), *Pdgfra^CreERT2/+^; Hgf^Δ/fl^* mice given tamoxifen at E8.5 have a muscleless diaphragm (*Figure 1M*) and partial or complete loss of muscle in limbs (*Figure 1N*), demonstrating a crucial role for HGF derived from PPF and limb fibroblasts. We also tested the alternative hypotheses that HGF is produced by myogenic cells and MET signaling is active in fibroblasts, but *Pax3^Cre/+^; Hgf^Δ/fl^* and *Pdgfra^CreERT2/+^; Met^Δ/fl^* mice have normal diaphragm and limb musculature (*Figure 1G, H, K and L*; controls in *Figure 1—figure supplement 3*). In summary, these data establish that HGF derived from PPF and limb fibroblasts induces MET signaling in somite-derived myogenic cells, which is required for muscularization of the diaphragm and the limbs.

### Diaphragm and shoulder muscle progenitors require fibroblast-derived *Hgf* during a similar temporal window

HGF/MET signaling is required for both diaphragm and limb muscle, but it is unclear whether *Hgf* is required during the same temporal window for development of diaphragm and forelimb muscles. This question is of particular interest because it has been proposed that a subset of shoulder muscle progenitors were recruited into the nascent PPFs during evolution, leading to the muscularization of the diaphragm in mammals (*Hirasawa and Kuratani, 2013*). If this were the case, shoulder muscle progenitors would be expected to migrate at a similar time and under the control of HGF/MET signaling as diaphragm progenitors in extant mammals. To dissect the temporal requirement for

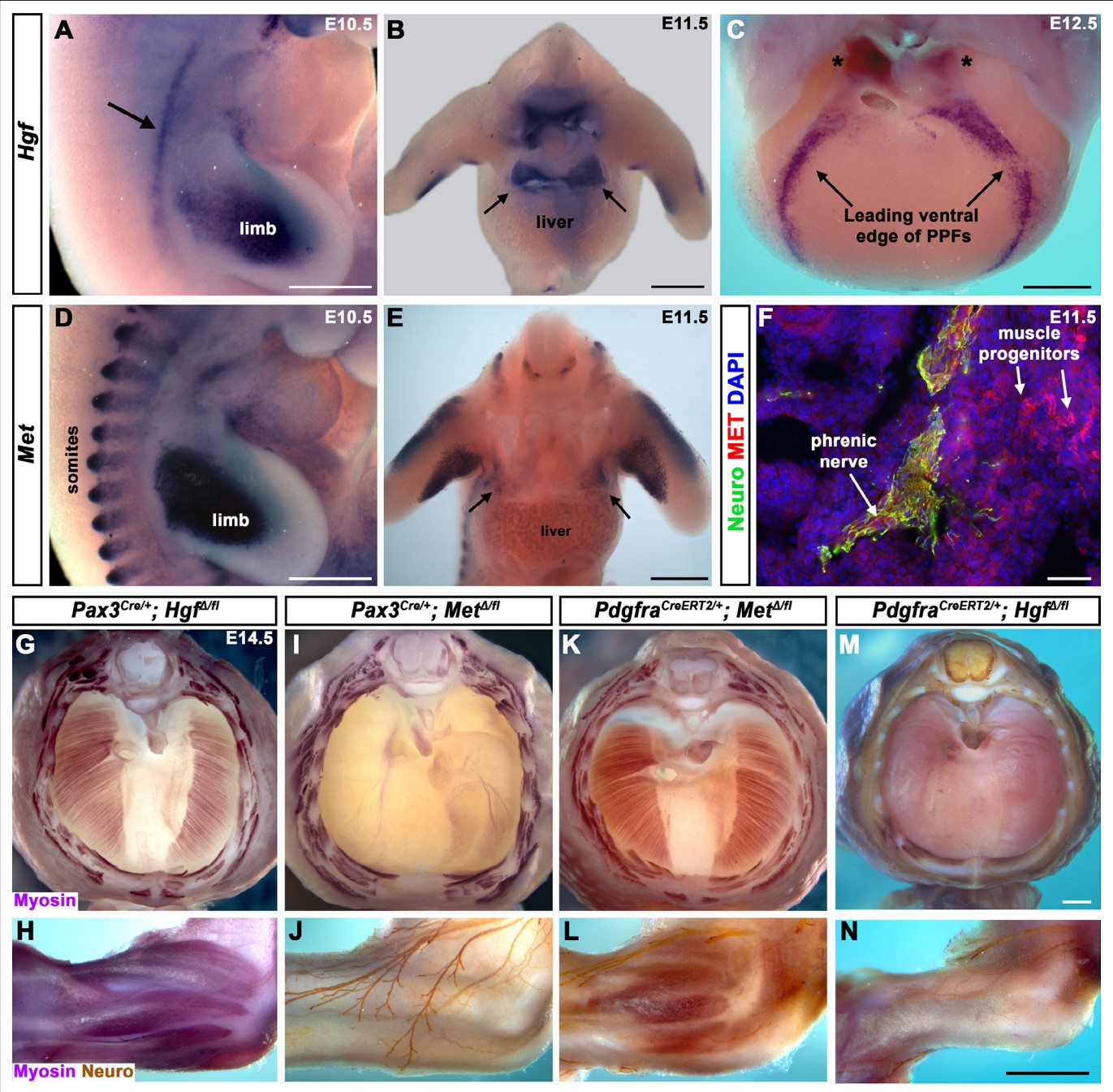

**Figure 1.** Fibroblast-derived *Hgf* and somite-derived *Met* are required for muscularization of the diaphragm and limb. (**A**) Lateral view at embryonic day (E) 10.5 of *Hgf* expression in lateral mesoderm adjacent to somites (arrow) and limb. (**B**) Cranial view of *Hgf* expression in E11.5 developing diaphragm (arrows) and limbs. (**C**) Cranial view of *Hgf* expression in E12.5 diaphragm at the leading edges of the pleuroperitoneal folds (PPFs) as they spread ventrally (arrows) and dorsally (asterisks). (**D**) Lateral view of E10.5 *Met* expression in muscle progenitors of limb and somites. (**E**) Cranial view of *Met* expression in E11.5 developing diaphragm (arrows) and limbs. (**A–E**) Expression via in situ hybridization. (**F**) MET and neurofilament immunofluorescence in transverse section through the phrenic nerve at E11.5. (**G, I, K, M**) E14.5 diaphragms stained for Myosin. (**H, J, L, N**) E14.5 forelimbs stained for Myosin and neurofilament. Deletion of *Met* in the *Pax3* lineage (**I, J**; n = 3/3) or *Hgf* in *Pdgfra* lineage (tamoxifen at E8.5) (**M, N**; n = 3/3) leads to muscleless diaphragms and muscleless or partially muscularized limbs. Conversely, deletion of *Hgf* in *Pax3* lineage (**G, H**; n = 3/3) or *Met* in *Pdgfra* lineage (tamoxifen at E9.5) (**K, L**; n = 3/3) results in normal diaphragm and limb muscle. Scale bars: (**A–E**) 500 µm; (**F**) 50 µm; (**G, I, K, M**) 500 µm; (**H, J, L, N**) 500 µm.

The online version of this article includes the following source data and figure supplement(s) for figure 1:

**Source data 1.** Limb and diaphragm phenotypes at embryonic day (E) 14.5 following deletion of *Hgf* and *Met* in *Pax3* and *Pdgfra* lineages.

*Figure 1 continued on next page*

*Figure 1 continued*

**Figure supplement 1.** *Hgf* and *Met* expression in the embryonic diaphragm and MET+ muscle progenitors are absent in *Met*^Δ/Δ mutant.

**Figure supplement 1—source data 1.** qPCR for *Pax7*, *Met*, and *Hgf* from embryonic day (E) 12.5 pleuroperitoneal folds (PPFs).

**Figure supplement 2.** *Pdgfra* is expressed within the developing diaphragm and *Pdgfra*^CreERT2 allele targets the pleuroperitoneal folds (PPFs).

**Figure supplement 3.** Control genotypes for embryos in *Figure 1I–N*.

fibroblast-derived *Hgf* for diaphragm and forelimb muscles, we examined these muscles in E16.5–18.5 *Pdgfra*^CreERT2/+; *Hgf*^Δ/flox mice given tamoxifen at E9.5 or E10.5, when muscle progenitors are actively delaminating from the somites and migrating into the nascent diaphragm and forelimb. We initially gave 6 mg of tamoxifen at E9.5, but only two embryos survived. Based on these two embryos, there was no obvious difference between 6 mg versus 3 mg tamoxifen on muscle: one embryo given 6 mg had muscleless limbs and diaphragm (*Figure 2Q–T*), while the other had partial muscle in the diaphragm with normal limb muscle. Therefore, we used 3 mg of tamoxifen for all subsequent experiments and compared diaphragm and limb defects in individual embryos (each row in *Figure 2* shows diaphragm and forelimb muscle from a single embryo). In the most mildly affected embryos, the diaphragm is missing a small ventral patch of muscle with normal forelimb muscles (*Figure 2E–H*; n = 3/10, compare with control in *Figure 2A–D*). In the most severely affected case, both the forelimb and diaphragm are muscleless (n = 1/10). A small number of mutants had muscleless diaphragms, but normal forelimb muscles (n = 2/10). Strikingly, a subset of mutants had muscleless (or nearly muscleless) diaphragms and displayed specific defects in shoulder musculature (*Figure 2I–P*; n = 4/10). The acromiodeltoid was absent or reduced and mispatterned (*Figure 2L and P*) and the spinodeltoid was strongly reduced in size, while other forelimb muscles appeared normal (*Figure 2K and O*). In all cases, the body wall muscles developed normally (e.g., *Figure 2Q*). When tamoxifen was given at E10.5, diaphragms had partial muscle, with normal forelimb muscles (n = 3/3; data not shown). In summary, muscleless limbs are always accompanied by a muscleless diaphragm, suggesting that these embryos had an early defect whereby muscle progenitors were unable to delaminate from the somites and migrate into nascent forelimbs and diaphragm. Partially muscularized diaphragms are associated with normal limb muscle. While a muscleless or nearly muscleless diaphragm may or may not have accompanying limb defects, loss of shoulder muscle was always associated with a muscleless or nearly muscleless diaphragm. These intermediate phenotypes indicate that most muscle progenitors migrate into the forelimb in advance of progenitors migrating into the diaphragm. However, based on their similar temporal sensitivity to HGF/MET signaling, the shoulder acromiodeltoid and spinodeltoid progenitors migrate at a similar time as the diaphragm progenitors. Thus continued expression of *Hgf* at this later developmental time point is required for recruitment of muscle cells necessary for development of diaphragm and shoulder muscles.

## Muscle formation, requiring PPF-derived HGF, controls phrenic nerve defasciculation

HGF signaling can act as a neurotrophic factor and chemoattractant in spinal motor neurons and cranial axons (*Caton et al., 2000*; *Ebens et al., 1996*; *Isabella et al., 2020*). However, the function of HGF and MET in the development of the phrenic nerve, the sole source of motor innervation in the diaphragm, has not been examined. To test the role of HGF/MET signaling in the phrenic nerve, *Hgf*^Δ/Δ, *Met*^Δ/Δ, and *Prrx1Cre*^Tg/+;*Hgf* ^fl/fl (Tg, transgene) mice were stained for neurofilament (*Figure 3*). By E12.0, in control mice the phrenic nerve has reached the PPFs and defasciculates into numerous small branchpoints prior to the full extension of the three primary branches (*Figure 3A*, arrows). However, in *Hgf*^Δ/Δ mutants, while the phrenic nerves reach the surface of the diaphragm, they do not correctly branch and defasciculate (*Figure 3B*). Instead of arborizing into numerous small branches, the right phrenic nerve bifurcates around the vena cava (n = 3/3; *Figure 3B*) and the left phrenic nerve fails to defasciculate to the same extent as in control embryos (*Figure 3B*). To test whether the PPF fibroblasts are a critical source of *Hgf* for branching of the phrenic nerve, *Hgf* was conditionally deleted using *Prrx1Cre*^Tg (*Logan et al., 2002*), which robustly recombines in PPF-derived fibroblasts (*Merrell et al., 2015*). Consistent with *Pdgfra*^CreERT2/+;*Hgf*^Δ/fl mice, the diaphragms of *Prrx1Cre*^Tg/+;*Hgf*^Δ/fl are muscleless (n = 5/12) or partially muscularized (n = 7/12). While the phrenic nerves reach the muscleless diaphragm, they lack primary and secondary branches (*Figure 3C and D*, arrows). Thus, loss of PPF-derived HGF leads to both muscle defects and phrenic nerve

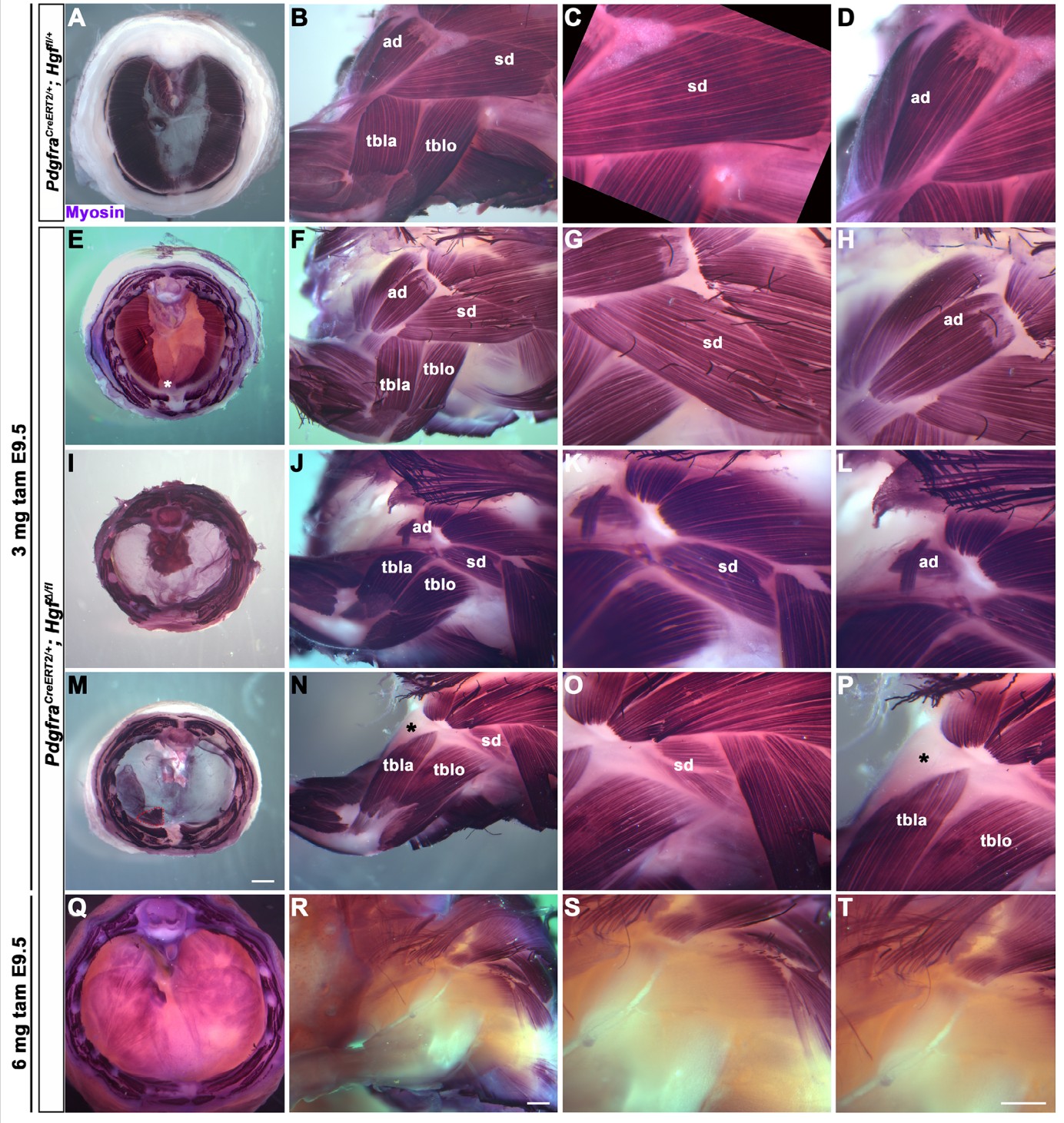

**Figure 2.** Reduced and mispatterned acromiodeltoid and spinodeltoid accompanies loss of diaphragm muscle following deletion of *Hgf* in the *Pdgfra* lineage. (**A–D**) Diaphragm and limb musculature in control *Pdgfra*^CreERT2/+^; *Hgf* ^fl/+^ given 3 mg tamoxifen at embryonic day (E) 9.5. (**E–H, I–L, M–P**) Diaphragm and limb muscle in three mutant *Pdgfra*^CreERT2/+^; *Hgf*^fl/fl^ embryos given 3 mg tamoxifen at E9.5. In the mildest phenotype, loss of ventral diaphragm muscle (asterisk, **E**), but normal limb and shoulder muscles (**F–H**; n = 3/10). In the moderate phenotype, shoulder muscles were affected by the diaphragm (n = 4/10). Absence of diaphragm muscle (**I**) accompanied by reduced spinodeltoid and mispatterned acromiodeltoid (**J–L**). Similarly, loss of diaphragm muscle, except in ventral-most region (red dotted line, remaining purple is AP stain trapped in connective tissue layer, **M**) and normal limb muscle except spinodeltoid reduced and acromiodeltoid absent (asterisk, **N–P**). (**Q–T**) In *Pdgfra*^CreERT2/+^; *Hgf*^fl/fl^ embryo given 6 mg tamoxifen at E9.5 near complete loss of both diaphragm and limb muscle (n = 1/2). Embryos harvested between E16.5 and E18.5. All samples stained with Myosin antibody. ad,

*Figure 2 continued on next page*

Figure 2 continued

acromiodeltoid; sd, spinodeltoid; tbla, triceps brachii lateral; tblo, triceps brachii long. Scale bars: (**A, E, I, M**) 1 mm; (**B, F, J, N, Q, R**) 500 μm; (**C, D, G, H, K, L, O, P, S, T**) 500 μm.

The online version of this article includes the following source data for figure 2:

**Source data 1.** Limb and diaphragm phenotypes at embryonic days (E) 16.5–E18.5 following deletion of *Hgf* in *Pdgfra* lineage.

defasciculation defects. Similar defasciculation phenotypes are also present in *Met*$^{Δ/Δ}$ mutants. Confocal analysis of E11.5 diaphragms, when the phrenic nerve is just reaching the PPFs, reveals that defasciculation defects are present in *Met*$^{Δ/Δ}$ mutants by this early time point (***Figure 3I and J***). Comparison of E11.5 *Met*$^{+/+}$, *Met*$^{Δ/+}$, and *Met*$^{Δ/Δ}$ diaphragms reveals a dose-dependent requirement for *Met* as the number of

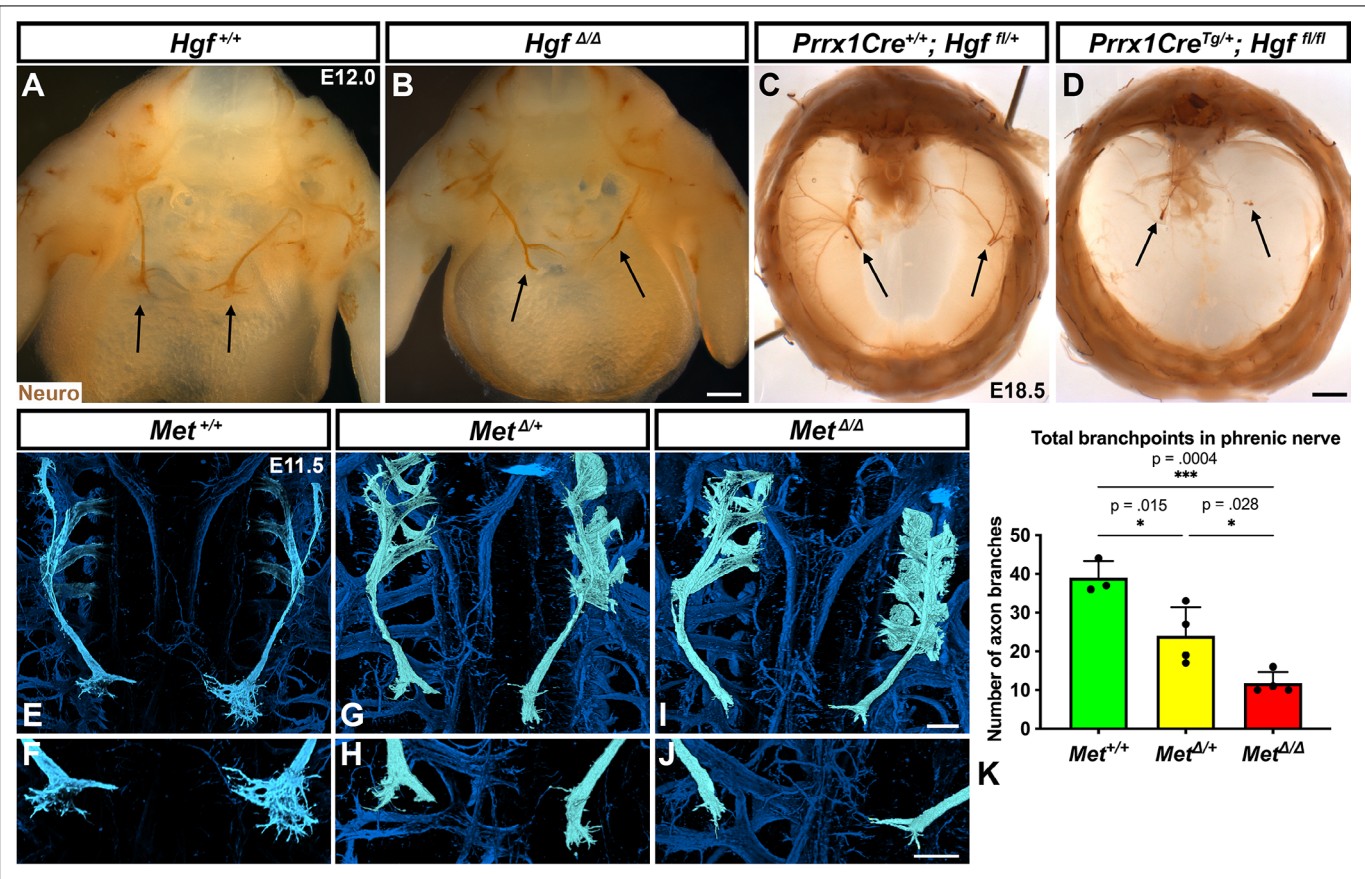

**Figure 3.** Loss of *Hgf* and *Met* leads to defasciculation defects in the phrenic nerve. Whole-mount neurofilament staining of the phrenic nerve in control (**A, C, E, F**) or *Hgf* (**B, D**) or *Met* mutants (**G–J**). Cranial view of dissected diaphragm region viewed with light microscopy (**A–D**) showing loss of phrenic nerve branches in *Hgf*$^{Δ/Δ}$ (n = 3) diaphragm (arrows, **A, B**) and *Prrx1Cre*$^{Tg/+}$; *Hgf*$^{fl/fl}$ diaphragm (arrows, **C, D**). Dorsal whole-mount view via confocal microscopy shows reduced phrenic nerve defasciculation in *Met*$^{Δ/+}$ and *Met*$^{Δ/Δ}$ diaphragms (**E–J**). Phrenic nerve and C3-5 spinal nerves pseudocolored in light blue. (**K**) Quantification of phrenic nerve branchpoints at embryonic day (E) 11.5 in *Met*$^{+/+}$ (n = 3), *Met*$^{Δ/+}$ (n = 4), and Met$^{Δ/Δ}$ (n = 4) embryos. Significance tested with one-way ANOVA; error bars represent standard error of the mean (SEM). Scale bars: (**A, B**) 250 μm; (**C, D**) 1 mm; (**E, G, I**) 100 μm; (**F, H, J**) 100 μm.

The online version of this article includes the following source data and figure supplement(s) for figure 3:

**Source data 1.** Diaphragm and phrenic nerve phenotypes following deletion of *Hgf* or *Met*.

**Figure supplement 1.** At embryonic day (E) 11.5, limb length and crown-rump length were not significantly different between *Met*$^{+/+}$, *Met*$^{Δ/+}$, and *Met*$^{Δ/Δ}$ embryos; branchpoints in *Met*$^{Δ/+}$ embryos are equivalent to *Met*$^{+/+}$ at E12.5.

**Figure supplement 1—source data 1.** qPCR for *Pax7*, *Met*, and *Hgf* from embryonic day (E) 12.5 pleuroperitoneal folds (PPFs).

**Figure supplement 2.** Loss of *Met* in Olig2 lineage does not lead to defects in phrenic nerve branching and *Hgf* is not sufficient to rescue defasciculation defects in the absence of diaphragm muscle.

**Figure supplement 2—source data 1.** Number of axon branchpoints after deletion of *Met* in *Olig2* lineage and in *Pax3*$^{SpD/SpD}$ mutants.

fascicles is lower in heterozygotes and is further reduced in homozygous mutants (*Figure 3E–K*). Importantly, the reduced number of branches in *Met* $^{\Delta/+}$ nerves indicates that the reduced branching is not merely the result of the total loss of muscle, as *Met* $^{\Delta/+}$ embryos have normally muscularized diaphragms (e.g., see *Figure 1K*). One potential cause of the reduced defasciculation defect in *Met* $^{\Delta/+}$ embryos is developmental delay. Based on crown rump length and limb length, however, *Met* $^{\Delta/+}$ embryos are not developmentally delayed relative to *Met* $^{+/+}$ embryos at E11.5 (*Figure 3—figure supplement 1A–E*). We also tested whether reduced defasciculation in *Met* $^{\Delta/+}$ embryos persists at later time points, but we found it resolves by E12.5 (*Figure 3—figure supplement 1F–I*).

To test for the requirement of MET within the phrenic nerve, motor neuron-specific deletion of *Met* was performed using *Olig2* $^{Cre}$ (*Zawadzka et al., 2010*). However, defasciculation defects were not present at E11.5 in the phrenic nerve (*Figure 3—figure supplement 2A–C*). These data suggest that MET does not intrinsically regulate phrenic nerve branching, but instead PPF-derived HGF may regulate phrenic nerve branching indirectly via muscle. To test this, we analyzed at E11.5 the diaphragms of *Pax3* $^{SpD/SpD}$ embryos (*Vogan et al., 1993*), which are muscleless, but maintain *Hgf* expression (*Figure 3—figure supplement 2G and H*; Merrell et al., 2015). In *Pax3* $^{SpD/SpD}$ diaphragms, axon branchpoints are strongly reduced (*Figure 3—figure supplement 2D–F*), similar to *Met* $^{\Delta/\Delta}$ mutants. Thus, HGF in the absence of muscle is not sufficient to promote normal phrenic nerve defasciculation. Altogether these data demonstrate that PPF-derived HGF, via muscle, is required for normal phrenic nerve defasciculation and primary branching.

## *Hgf* is required in fibroblasts to fully muscularize the diaphragm after delamination of muscle progenitors from somites

While HGF/MET signaling is critical for delamination of muscle progenitors from the somites (*Dietrich et al., 1999*), it is unclear whether HGF plays a later role in development of the diaphragm's muscle. To test the later temporal requirement of HGF in PPF fibroblasts, we deleted *Hgf* via *Pdgfra* $^{CreERT2/+}$; *Hgf* $^{\Delta/fl}$ mice given tamoxifen at different time points. When *Pdgfra* $^{CreERT2/+}$; *Hgf* $^{\Delta/fl}$ mice were given tamoxifen at E9.0, prior to the onset of muscle precursor migration to the diaphragm (*Sefton et al., 2018*), the diaphragm lacks all muscle (*Figure 4B*; n = 3/3). This is likely due to a failure of muscle progenitors to delaminate and emigrate from the somites, as in *Met-null* diaphragms (*Dietrich et al., 1999*). When *Hgf* is deleted via tamoxifen at E9.5, when muscle progenitors are delaminating and migrating to the nascent diaphragm (*Sefton et al., 2018*), the diaphragm displays large ventral muscleless regions as well as dorsal muscleless patches at E14.5 (*Figure 4C*; n = 6/6). Notably, the phrenic nerves in these diaphragms only extend to the regions with muscle (*Figure 4H*). When *Hgf* is deleted via tamoxifen at E10.5, the muscle reaches its normal ventral extent in most E14.5 embryos (*Figure 4D*, n = 11/13). However, when these embryos are allowed to develop to E17.5 (when the muscle has normally expanded to the ventral midline), a large ventral muscleless region persists in mutant embryos (*Figure 4F*; n = 3/3). When mutants are given tamoxifen at E11.5, after migration of progenitors to the PPFs has completed (*Sefton et al., 2018*), a smaller muscleless region is present in the ventral diaphragm at E17.5 (*Figure 4G*; n = 4/4). Thus, these data demonstrate that after its initial requirement for muscle precursor delamination from the somites, PPF-derived *Hgf* is critical for muscularization of the ventral- and dorsal-most regions of the diaphragm. This role for HGF is consistent with its strong expression in these ventral- and dorsal-most regions (*Figure 1C*).

We next sought to determine how PPF-derived HGF regulates development of the dorsal and ventral-most regions of the diaphragm muscle. Our previous studies (*Merrell et al., 2015*; *Sefton et al., 2018*) have shown that the PPFs expand dorsally and ventrally, carrying muscle as they expand, and therefore control overall morphogenesis of the diaphragm. Absence of dorsal and ventral muscle regions in *Hgf* mutants could result from a failure of the PPFs to expand dorsally and ventrally and thus lead to the consequent lack of dorsal and ventral diaphragm muscle. To test whether PPF expansion is aberrant following deletion of *Hgf*, we examined *Prrx1Cre* $^{Tg/+}$; *Hgf* $^{\Delta/fl}$; *Rosa26* $^{LacZ/+}$ mice, in which we genetically labeled PPFs as they spread across the surface of the liver at E13.5. However, the PPFs reach their normal ventral extent at E13.5 following loss of *Hgf* (*Figure 4—figure supplement 1A–C*). To examine whether fibroblasts populate muscleless regions following deletion of *Hgf*, we stained for Pax7, MyoD, Myosin, and GFP in *Pdgfra* $^{CreERT2/+}$; *Hgf* $^{\Delta/fl}$; *Rosa26* $^{mTmG/+}$ mice at E15.5. GFP+ fibroblasts were present throughout large muscleless regions (*Figure 4—figure supplement 1D–F*). These data argue that the loss of ventral and dorsal muscle is not due to defects in PPF morphogenesis or

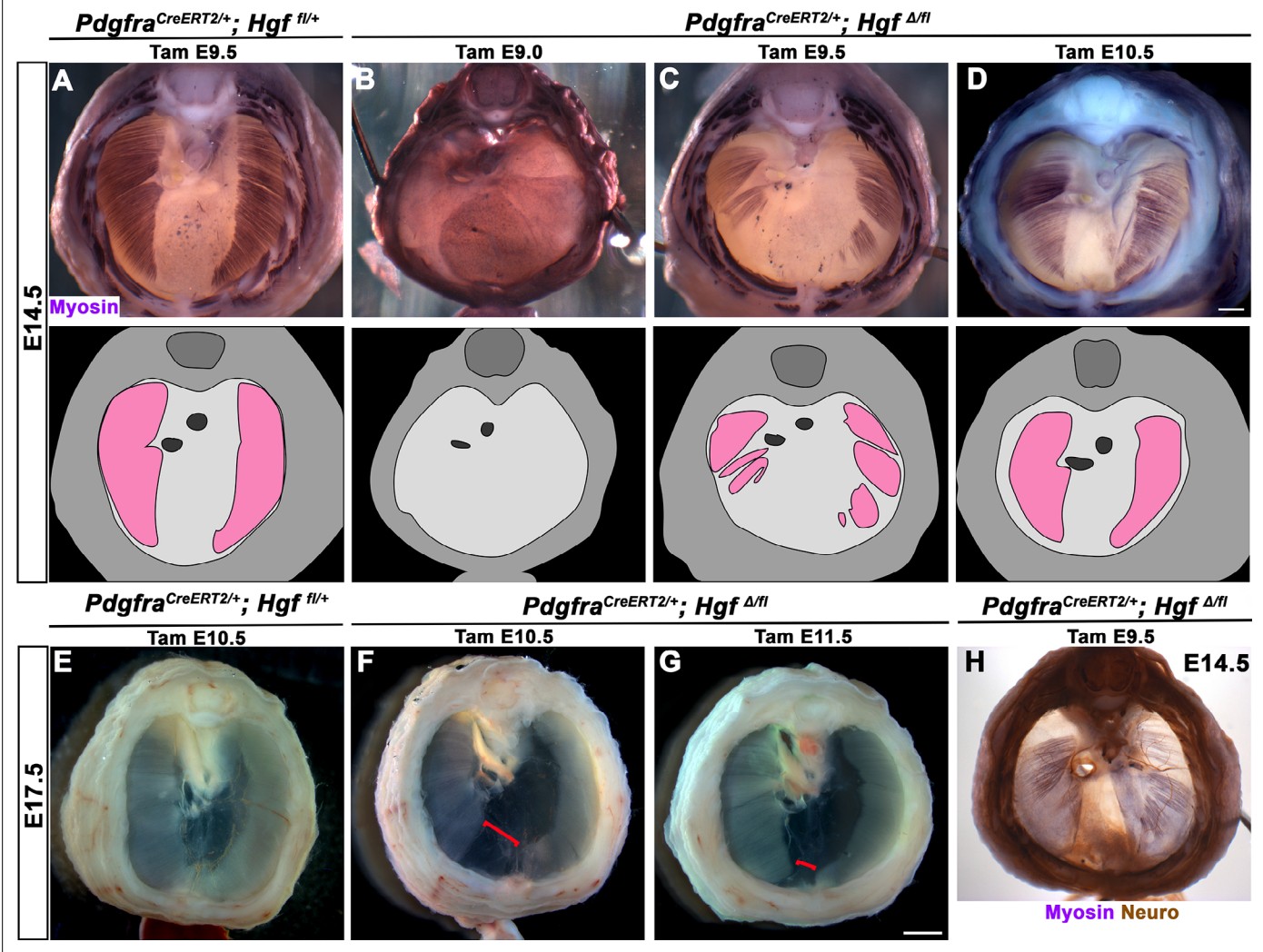

**Figure 4.** Loss of diaphragm muscle following timed deletion of *Hgf* in the pleuroperitoneal fold (PPF) fibroblast lineage. (**A–D**) Upper panels: cranial view of embryonic day (E) 14.5 diaphragms stained for Myosin. Middle row panels: illustrations of muscle distribution. (**A**) Control *Pdgfra^CreERT2/+^; Hgf^fl/+^* diaphragm muscle forms two lateral wings that have not yet converged ventrally. (**B**) Muscleless diaphragm in *Pdgfra^CreERT2/+^; Hgf^Δ/flox^* when tamoxifen is administered at E9.0 (n = 3/3). (**C**) Large muscleless regions in ventral and dorsal diaphragm following tamoxifen administration at E9.5 (n = 6/6). (**D**) *Pdgfra^CreERT2/+^; Hgf^Δ/flox^* muscle reaches normal ventral extent when tamoxifen is administered at E10.5 (n = 11/13). (**E–G**) Unstained E17.5 diaphragms in cranial view. (**E**) Control diaphragm muscle has closed ventrally. (**F**) Large ventral muscleless region following *HGF* deletion at E10.5 (red bracket, n = 3/3). (**G**) Smaller ventral muscleless region following tamoxifen at E11.5 (red bracket, n = 4/4). (**H**) Cranial view of E14.5 *Pdgfra^CreERT2/+^; Hgf^Δ/flox^* mutant with tamoxifen administered at E9.5. Diaphragm stained for Myosin and neurofilament, indicating phrenic nerve tracks with regions of muscle (n = 3/3). Scale bars: (**A–D, H**) 500 µm; (**E–G**) 1 mm.

The online version of this article includes the following source data and figure supplement(s) for figure 4:

**Source data 1.** Timed deletion of *Hgf* in the pleuroperitoneal fold (PPF) fibroblast lineage.

**Figure supplement 1.** Pleuroperitoneal folds (PPFs) spread normally and PPF fibroblasts are present in muscleless regions following deletion of *Hgf*.

**Figure supplement 1—source data 1.** Sample information for spreading of pleuroperitoneal folds (PPFs) after deletion of *Hgf*.

survival of fibroblasts. Moreover, PPF expansion is not dependent on HGF or muscularization of the diaphragm.

## Development of dorsal and ventral regions of diaphragm muscle requires continuous MET signaling

Our experiments conditionally deleting *Hgf* after emigration of myogenic progenitors from the somites indicate that HGF/MET signaling plays additional later roles in the development of the diaphragm's

muscle. To specifically test when MET signaling is required in myogenic cells, we first deleted *Met* using *Pax7^iCre/+* or tamoxifen-inducible *Pax7^CreERT2* mice (*Keller et al., 2004*; *Murphy et al., 2011*), which cause Cre-mediated recombination later than *Pax3^Cre* in a subset of embryonic myogenic progenitors as well as all fetal and adult progenitors (*Hutcheson et al., 2009*). Neither *Pax7^iCre/+*; *Met^Δ/fl* nor *Pax7^CreERT2/+*; *Met^Δ/fl* embryos displayed any defects in diaphragm muscularization at E14.5 or P0 (*Figure 5—figure supplement 1*). This may indicate that *Met* is not required during fetal myogenesis as *Pax7* is primarily expressed in fetal myogenic progenitors. However, *Met* derived from embryonic *Pax3+Pax7-* myogenic progenitors is likely present in the muscle of these mutant diaphragms and so does not permit analysis of the consequence of *Met* deletion in muscle.

As an alternate strategy to test when MET signaling is required, we turned to an ATP-competitive inhibitor of MET autophosphorylation, BMS777607 (as well as MET-related kinases RON and AXL; *Schroeder et al., 2009*), which inhibits phospho-Met in muscle progenitors (*Figure 6—figure*

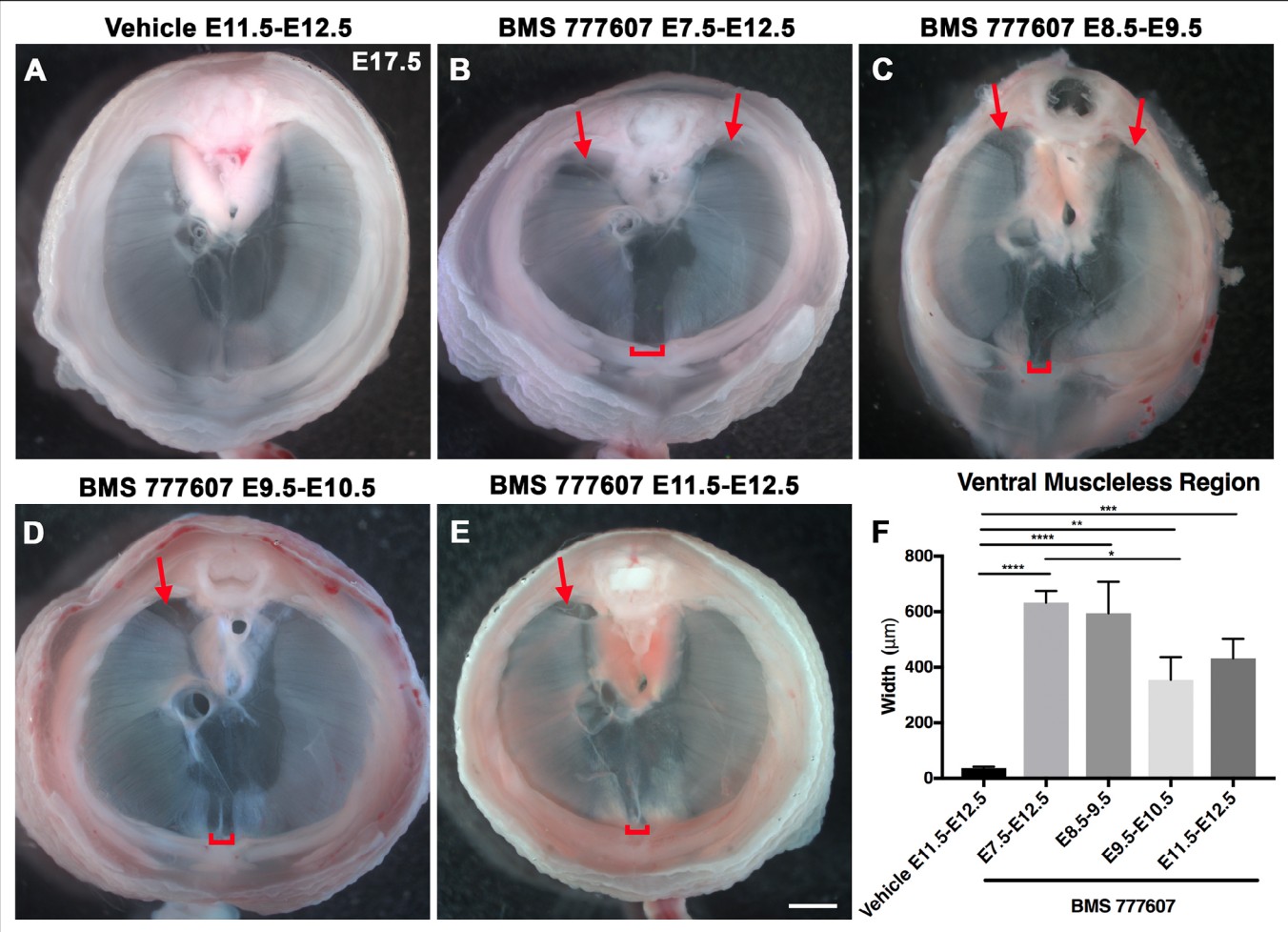

**Figure 5.** Reduction of Met signaling through inhibitor BMS777607 results in muscleless dorsal and ventral regions of the diaphragm. (**A-E**) Unstained E17.5 diaphragms in cranial view. (**A**) The left and right portions of the costal diaphragm meet in the ventral midline by embryonic day (E) 17.5 in vehicle-treated controls (n = 12/12). (**B–E**) Dorsal left, dorsal right (arrows), and ventral midline regions (brackets) are muscleless when treated with BMS777607 daily between E7.5 and E12.5 (**B**: n = 7; **C**: n = 6; **D**: n = 8; **E**: n = 9). (**F**) Width of ventral midline muscleless region is significantly larger than vehicle-treated controls when BMS777607 is administered at either early (E7.5–E8.5) or at later stages of diaphragm development (E11.5–E12.5). Significance tested with one-way ANOVA. *p<0.05, **p<0.01, ***p<0.001, ****p<0.0001. Error bars represent standard error of the mean (SEM). Scale bars (**A–E**) 1 mm.

The online version of this article includes the following source data and figure supplement(s) for figure 5:

**Source data 1.** Measurements of ventral muscle gap after timed treatments with BMS777607.

**Figure supplement 1.** Normal muscularization of the diaphragm following deletion of *Met* in the *Pax7* lineage.

**Figure supplement 1—source data 1.** Sample information for *Met* deletion in *Pax7* lineage.

*supplement 1A*). We administered daily doses of BMS777607 to pregnant females to temporally inhibit MET signaling. In vehicle-treated controls harvested at E17.5, the diaphragm is completely muscularized (n = 12/12; *Figure 5A*). When BMS777607 was administered daily between E7.5 and E12.5, all embryos (n = 7/7) displayed bilateral dorsal muscleless patches and a ventral muscleless region (*Figure 5B*, arrows). When treated at E8.5–E9.5 or E9.5–10.5 (*Figure 5C and D*), all diaphragms had ventral muscleless regions (n = 14/14) and 35% had dorsal muscleless regions (n = 5/14). When treated at E11.5 and E12.5, after diaphragm progenitors have fully delaminated from the somites (*Sefton et al., 2018*), embryos had ventral muscleless regions (*Figure 5E and F*; n = 8/8) and dorsal muscleless regions (n = 3/8). Quantification of the size of the ventral muscleless region indicates that all MET inhibition strategies lead to muscleless regions, with the largest muscleless regions when MET is inhibited E7.5–E12.5 or E8.5–E9.5 (*Figure 5F*). These data demonstrate that MET is continuously required from E7.5 to E12.5 for complete muscularization of the diaphragm. The regions requiring continuous MET signaling are on the leading edges of the diaphragm: the bilateral dorsal muscle and ventral midline muscle. These are the last regions to receive muscle progenitors that differentiate into myofibers. For both *Hgf* deletion in fibroblasts and global MET inhibition, loss during muscle migration from somites at approximately E9.5 leads to dorsal and ventral muscleless regions, while later loss at E11.5 leads to primarily ventral muscleless regions. The ventral midline of the diaphragm does not fully close until E16.5, likely making it more susceptible to later perturbations.

## MET signaling is required for survival and motility of diaphragm muscle progenitors in vitro

While the regions most sensitive to MET inhibition are those that differentiate latest, it is unclear whether MET is required for proliferation, survival, differentiation, and/or motility. To investigate the function of MET in diaphragm muscle progenitors, we turned to an in vitro system to co-culture E12.5 diaphragm fibroblasts and myoblasts (*Bogenschutz et al., 2020*) in combination with BMS777607. PPF explants were dissected from E12.5 *Pax3^{Cre/+}; Rosa26^{nTnG/+}* embryos (*Engleka et al., 2005*; *Prigge et al., 2013*), in which *Pax3*-derived myogenic nuclei are GFP+ and PPF fibroblast nuclei are Tomato+, and cultured them with either 10 µM BMS777607 or DMSO vehicle control (*Figure 6—figure supplement 2*). Overall, growth of GFP+ muscle progenitors was impaired with inhibitor treatment (*Figure 6A–C*). To assess effects of the inhibitor on the number of myoblasts, we examined *MyoD*. After 48 hr in culture, *MyoD* expression was reduced with inhibitor treatment (*Figure 6D*), and the percentage of cells co-expressing GFP and MyoD protein was similarly abrogated (*Figure 6E and F*). By contrast, expression of the PPF fibroblast marker *Gata4* was not significantly changed following inhibitor treatment (*Figure 6G*). We tested whether the decreased growth of myogenic cells was due to decreased proliferation or increased apoptosis. Analysis of GFP+ myogenic cells labeled via EdU indicates that BMS777607 treatment does not significantly change the percentage of proliferating cells (*Figure 6H and I*). However, examination of apoptotic cells via staining for cleaved Caspase-3 showed that BMS777607 treatment significantly increased the number of apoptotic GFP+ myogenic cells (*Figure 6J and K*). To examine the relevant pathway(s) for increased apoptosis, we assayed the expression of *Fas* (a cell surface death receptor), tumor protein *Trp53* (which encodes p53), and autophagy marker *Map1l3ca* in BMS777607 and vehicle-treated PPFs. Both *Fas* and *Map1l3ca* are significantly upregulated following treatment with BMS777607, while *Trp53* is unaffected (*Figure 6—figure supplement 1B–D*). Inhibition of MET is also known to impair cell motility (reviewed by *Birchmeier et al., 2003*), and we found that the motility of GFP+ cells treated with BMS777607 was impaired, with reduced velocity and lower overall displacement (*Figure 6L–M*). We also assessed effects on cell morphology by examining diaphragm muscle progenitors, labeled with membrane-bound GFP (via *Pax3^{Cre/+}; Rosa26^{mTmG/+}*). Cells were significantly more circular with BMS777607 treatment, which is consistent with compromised survival and motility (*Figure 6N and O*). Overall, these data show that MET signaling is important for survival and motility of diaphragm muscle progenitors in vitro.

## MET signaling is required for the population of muscle progenitors at the diaphragm's leading edges and the consequent development of the dorsal and ventral-most muscle regions

Our in vivo studies show that HGF/MET signaling is required for the development of the dorsal and ventral-most regions of the diaphragm muscle, and our in vitro studies find that MET is required for

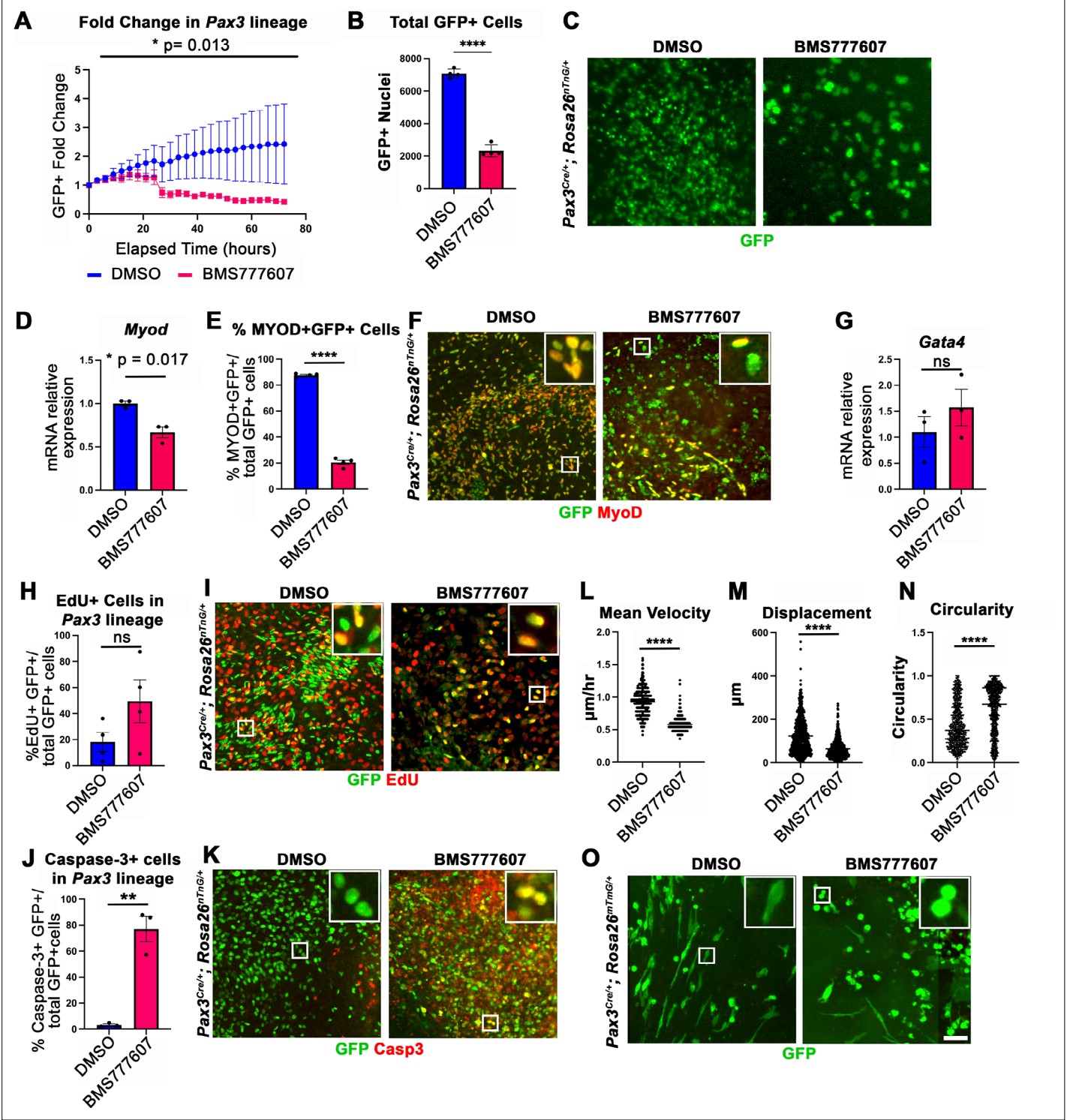

**Figure 6.** In vitro survival and motility of myogenic cells is impaired by pharmacological inhibition of Met signaling. (A–O) Embryonic day (E) 12.5 pleuroperitoneal folds (PPFs) isolated from *Pax3^{Cre/+}; Rosa26^{nTnG/+}* (A–M) or *Pax3^{Cre/+}; Rosa26^{mTmG/+}* (N, O) embryos cultured with DMSO vehicle control or BMS777607 and imaged on the ImageXpress Pico. (A–C) The average ratio of GFP+ fold change (cell number at time T/cell number at time 0) ± SEM is plotted (A). The fold change (A) and total final count (B) of GFP+ cells are reduced following treatment with BMS777607 after 72 hr in culture (n = 4 biological replicates). Representative GFP images shown in (C) at 72 hr in culture. (D–F) *MyoD* expression (via qPCR, n = 3, D) and percentage of MyoD+ GFP+ cells (n = 4) (E) is significantly lower after 48 hr in culture with MET signaling inhibition. Representative images of GFP and MyoD expression (F). (G) Expression of PPF fibroblast marker *Gata4* is not significantly affected by BMS777607 treatment (via qPCR, n = 3). (H, I) EdU labeling of GFP+ cells is not significantly changed by BMS777607 treatment (n = 4). (J, K) Cleaved Caspase-3 expression was significantly increased with BMS777607 treatment

*Figure 6 continued on next page*

*Figure 6 continued*

(n = 3). (**L, M**) Mean velocity and total displacement of peripheral GFP+ nuclei were decreased with BMS777607 treatment. GFP+ nuclei were imaged every 8 min over 14 hr to track cell motility (>500 nuclei measured from n = 3). (**N, O**) BMS777607-treated GFP+ cells were significantly more circular (>1000 cells measured from n = 3). Representative images (**O**). *p<0.05, **p<0.01, ****p<0.0001. Statistical changes in cell number over time (**A**) were determined using repeated-measures ANOVA on the log2 transformed fold change over time. Statistical changes determined with unpaired t test in (B, D, E, G, H, L, M, N, J). Error bars represent standard error of the mean (SEM). Scale bars: (**C, F, I, K, O**) 100 μm.

The online version of this article includes the following source data and figure supplement(s) for figure 6:

**Source data 1.** In vitro effects on diaphragm muscle after pharmacological inhibition of MET signaling.

**Figure supplement 1.** BMS777607-mediated downregulation of phospho-Met and relevant pathways to cell survival in embryonic day (E) 12.5 diaphragms.

**Figure supplement 1—source data 1.** qPCR data for *Fas*, *Map1lc3a*, and *Trp53*.

**Figure supplement 2.** Initial muscle lineage and final nuclear density following vehicle and BMS777607 treatment.

muscle progenitor survival and motility. Based on these data, we hypothesized that in vivo loss of dorsal and ventral muscle regions is due to fewer muscle progenitors and/or myoblasts at the dorsal and ventral leading edges of the diaphragm when it is expanding. To test this, wild-type embryos were treated with BMS777607 daily E7.5–E11.5, harvested at E12.5, and stained for myogenic cells with a cocktail of PAX7, MyoD, and Myosin antibodies as well as for EdU, cleaved Caspase-3, and neurofilament. We found that the PPFs (identified and outlined in 3D by their unique morphology, viewed by autofluorescence) were more variable in size, but not significantly decreased in size from control diaphragms. We also found, consistent with our analysis of $Hgf^{\Delta/\Delta}$, and $Met^{\Delta/\Delta}$ mice, that nerve branching was strongly reduced by the inhibitor (*Figure 7C, F, J and M*; *Video 1*). Supporting our hypothesis, the inhibitor led to a reduction in the number of mononuclear progenitors and myoblasts at the ventral and dorsal leading edges of the muscle (*Figure 7A, D, H and K*, arrows; *Video 1*). Inhibitor-treated embryos also showed reduced numbers of EdU+ cells overall (*Figure 7B, E and G*) and an increased number of cleaved Caspase-3-positive cells within the PPFs (*Figure 7I, L and N*). To exclude early impacts of BMS777607 on muscle progenitor emigration from somites, BMS777607 or vehicle was also administered only E11.5–E12.5 to $Pax3^{Cre/+}$; $Rosa26^{mTmG/+}$ embryos, which were then harvested at E15.5. Although the number of mononuclear GFP+ cells were not significantly reduced, the number of Pax7/MyoD/Myosin-labeled cells was substantially reduced at the ventral leading edges with inhibitor treatment (*Figure 7O–V*). Thus, these data demonstrate that in vivo Met signaling is required to promote proliferation and survival of myogenic cells, and its inhibition leads to a loss of muscle progenitors and myoblasts at the leading edges of the PPFs (which express high levels of *Hgf* at E12.5) and results in a loss of the dorsal-most and ventral-most diaphragm muscle.

## Discussion

The diaphragm is an essential mammalian skeletal muscle, playing a critical role in respiration and serving as a barrier that separates the thorax from the abdomen. Not only is the diaphragm a functionally important muscle, but it serves as an excellent system to study muscle patterning and morphogenesis, since it is a flat muscle that largely develops in two dimensions. Development of the diaphragm, like other skeletal muscles, requires the integration of muscle, connective tissue, and nerve that arise from different embryonic sources. Our study establishes that the connective tissue fibroblasts are the source of a molecular signal, HGF, that directly controls the recruitment, survival, and expansion of MET+ muscle progenitors and indirectly, via muscle, regulates phrenic nerve branching (*Figure 8*).

Development of muscle and its innervating motor neurons must be tightly integrated to produce a functional muscle. The connective tissue is an ideal candidate tissue to orchestrate this process as it enwraps myofibers and neuromuscular junctions (*Nassari et al., 2017*; *Sefton and Kardon, 2019*). In the diaphragm, the PPFs are the source of the diaphragm's connective tissue fibroblasts and critical for overall diaphragm morphogenesis (*Merrell et al., 2015*). Thus, the PPFs are likely to coordinately regulate muscle and the phrenic nerve. A number of previous studies also suggested that HGF/MET is a key signaling pathway for this coordination since it has been found to regulate both muscle development and innervation (*Bladt et al., 1995*; *Dietrich et al., 1999*; *Ebens et al., 1996*; *Lamballe et al., 2011*; *Maina et al., 1996*; *Yamamoto et al., 1997*). In addition, previous studies established

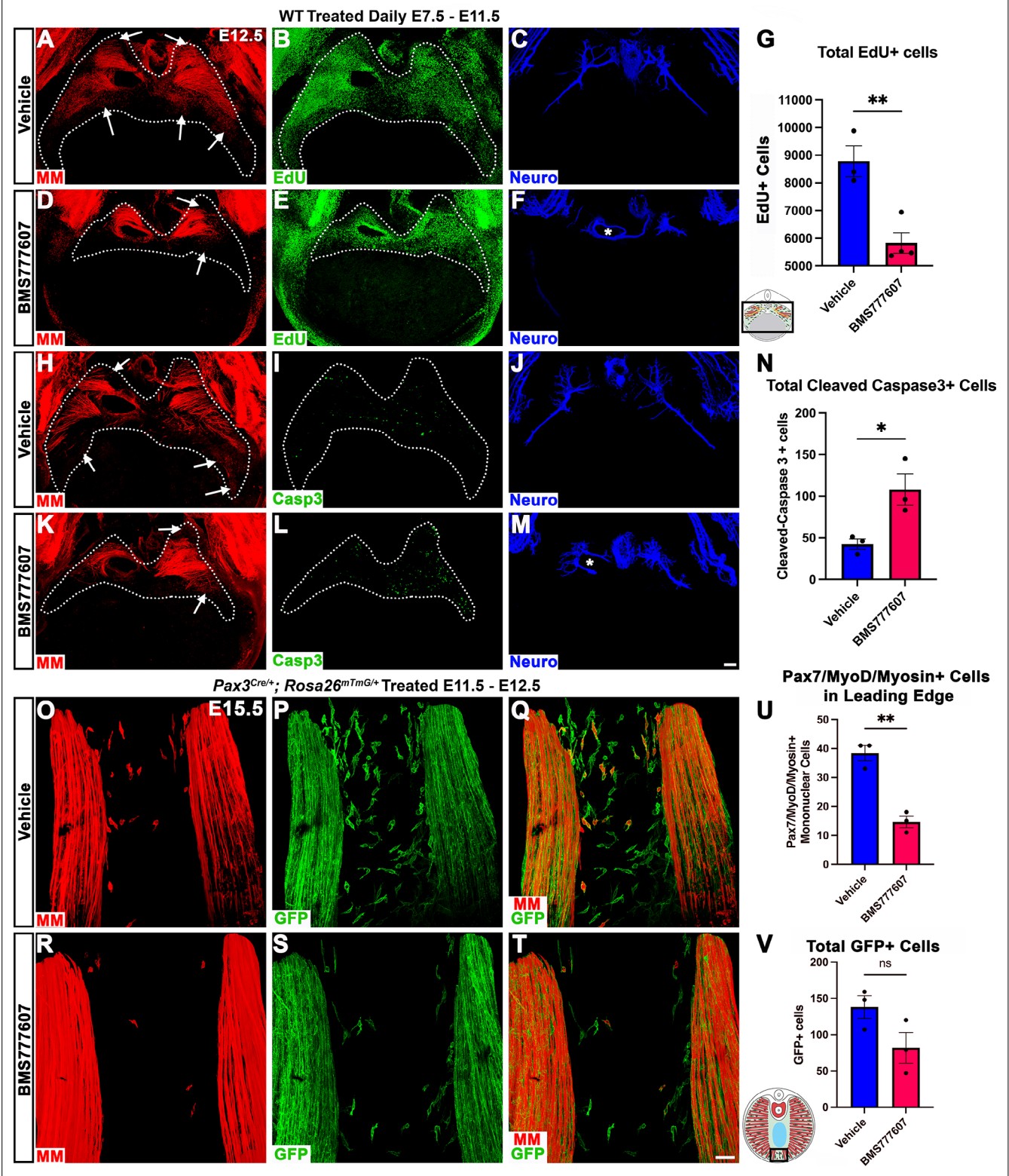

**Figure 7.** Inhibition of Met signaling in vivo alters cell proliferation, apoptosis, phrenic nerve morphology, and reduces muscle progenitors at the leading edge of the diaphragm. (**A–N**) WT embryos were treated with BMS777607 or vehicle control daily between embryonic day (E) 7.5 and E11.5, harvested at E12.5, stained for Pax7/MyoD/Myosin (MM) and neurofilament, and imaged in whole-mount cranial view on the confocal. Treatment with BMS777607 leads to fewer mononuclear myogenic cells on the dorsal and ventral leading edges of the diaphragm (arrows; **A, D, H, K**), reduced total EdU-positive nuclei (n = 3 vehicle treated; n = 4 BMS777607 treated; **B, E, G**), increased cleaved-caspase-3 positive cells (n = 3; **I, L, N**), and aberrant

*Figure 7 continued on next page*

*Figure 7 continued*

phrenic nerve branching (**F, M**), where the right phrenic nerve wraps around vena cava (asterisk in **F, M**). Schematic of region imaged (black box) in (**G**). (**O–V**) *Pax3^Cre/+^; Rosa26^mTmG/+^* embryos were treated with BMS777607 or vehicle at E11.5 and E12.5, harvested at E15.5 and stained for GFP and MM. Tomato is unlabeled. (**O–T**) Cranial view of leading ventral edges of the diaphragm at E15.5, with mononuclear muscle progenitors in region that will fill with muscle by E16.5. (**U**) Fewer mononuclear Pax7/MyoD/Myosin+ cells populate the leading edge following treatment with BMS777607 (n = 3). (**V**) Quantification of mononuclear GFP+ cells in vehicle or BMS777607-treated embryos. Significance analyzd with unpaired t test; error bars represent standard error of the mean (SEM). *p<0.05 **p<0.01.Schematic of region imaged (black box) in (**V**). Scale bars: (**A–F, H–M**) 100 µm; (**O–T**) 50 µm.

The online version of this article includes the following source data for figure 7:

**Source data 1.** In vivo reduction diaphragm muscle at the leading ventral edge after pharmacological inhibition of MET signaling.

that *Met* is required for diaphragm development (*Bladt et al., 1995*; *Dietrich et al., 1999*; *Maina et al., 1996*). Here, we used conditional mutagenesis to specifically target *Hgf* and *Met* in PPF-derived connective tissue fibroblasts or muscle progenitors to genetically dissect the role of these cells and HGF/MET signaling in diaphragm development (*Figure 8*). First, as expected we found that HGF/MET signaling is required for the initial delamination and migration of MET+ muscle progenitors from the somites to the nascent diaphragm. More surprisingly, we found that HGF expressed at the dorsal and ventral margins of the expanding PPFs is continuously required for proliferation, survival, and motility of muscle progenitors and the consequent expansion and full muscularization of the diaphragm. Our experiments also showed that, unlike the muscle progenitors, HGF/MET signaling is not required for the recruitment and targeting of phrenic nerve axons to the nascent diaphragm. However, PPF-derived HGF, through its regulation of muscle, is required for phrenic nerve defasciculation, primary branching, and subsequent branch outgrowth throughout the diaphragm muscle. Previous studies have identified how phrenic motor neurons are specified in the motor column (*Dasen et al., 2008*; *Dasen et al., 2003*; *Dasen et al., 2005*; *Jung et al., 2010*; *Rousso et al., 2008*) and later arborize (*Philippidou et al., 2012*) to form neuromuscular junctions (*Burden, 2011*; *Li et al., 2008*; *Wang et al., 2003*; *Yumoto et al., 2012*). Only one study (*Uetani et al., 2006*) has identified molecular regulators, Receptor Protein Tyrosine Phosphatases σ and δ, of phrenic nerve defasciculation and primary axon outgrowth. Altogether our study demonstrates that PPF-derived connective tissue fibroblasts and HGF directly control diaphragm muscle recruitment and expansion and indirectly, via muscle and the neurotrophic factors it expresses, control phrenic nerve branching and outgrowth. Connective tissue and HGF are also likely to regulate the development of other muscles and their motor neurons as connective tissue and HGF/MET signaling have been implicated in the development and innervation of limb and back muscles (*Caruso et al., 2014*; *Helmbacher, 2018*).

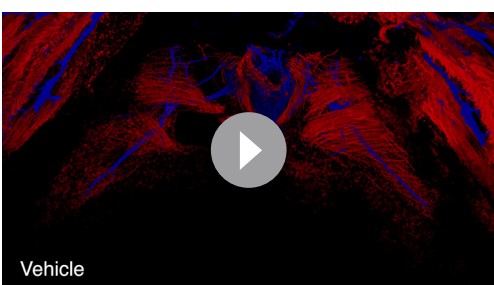

**Video 1.** Inhibition of Met signaling in vivo reduces muscle progenitors at the leading edges of the diaphragm. Embryos were treated with BMS777607 or vehicle control daily between embryonic day (E) 7.5 and E11.5, harvested at E12.5, stained for Pax7, MyoD, Myosin (muscle markers in red), and neurofilament (blue) and imaged in whole-mount cranial view on the confocal. Fewer mononuclear myogenic cells are present on the dorsal and ventral leading edges of the diaphragm at E12.5 and the phrenic nerve displays abnormal branching after treatment with BMS777607. https://elifesciences.org/articles/74592/figures#video1

Our study also provides insights into how the diaphragm might have evolved. The diaphragm muscle is unique to mammals and how it evolved in mammals is a major unanswered question (*Perry et al., 2010*). Evolution of the diaphragm involved the acquisition of developmental innovations in mammals that are absent from birds and reptiles. Comparison of these groups suggests the following important developmental innovations: formation and expansion of PPFs across the liver to separate the thoracic and abdominal cavities, recruitment of muscle progenitors to the PPFs and their expansion and differentiation into the radial array of myofibers, and recruitment and targeting of motor neurons to the diaphragm muscle (*Hirasawa et al., 2016*; *Hirasawa and Kuratani, 2013*; *Sefton et al., 2018*). HGF/MET signaling has important, conserved functions in the development of most vertebrate hypaxial muscles (*Adachi et al., 2018*; *Haines et al., 2004*; *Okamoto et al., 2019*). Here we identify that *Hgf* expressed by the PPFs is crucial for recruitment

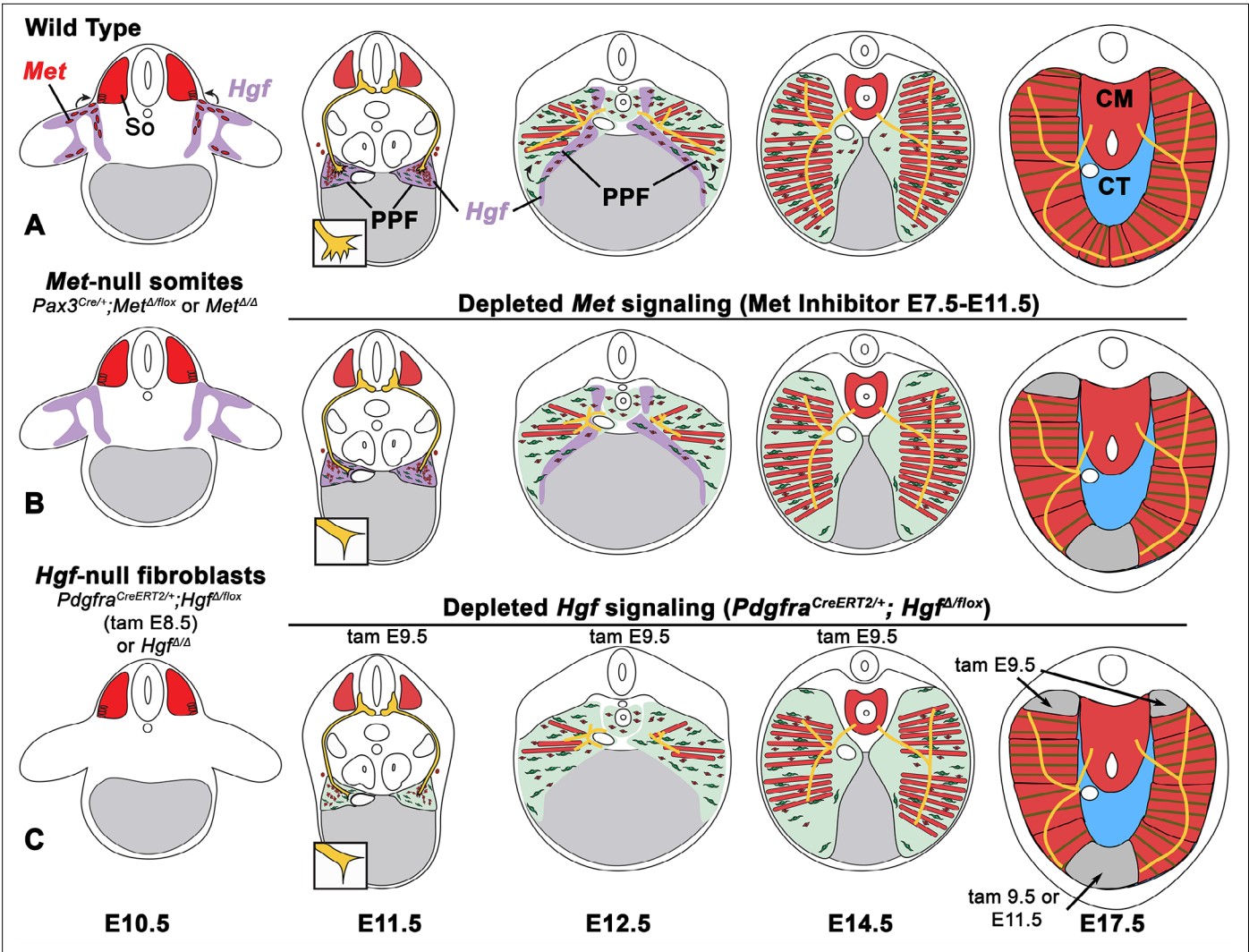

**Figure 8.** Model of HGF-MET signaling in skeletal muscle and phrenic nerve in the diaphragm. (**A**) Our data support a model where pleuroperitoneal fold (PPF) fibroblast-derived HGF directly regulates recruitment, survival, and expansion of MET+ muscle progenitors (in red) and indirectly, via muscle, regulates phrenic nerve (in yellow) branching and outgrowth. (**B**) *Met* null mutations or *Met* deletion in muscle progenitors (prior to embryonic day [E] 9.0) lead to muscleless limbs and diaphragm and reduced defasciculation of the phrenic nerve. Pharmacological inhibition of MET between E7.5 and E11.5 or later time points (i.e., E11.5–12.5) results in reduced muscle progenitors (via increased apoptosis and reduced motility) at the diaphragm's leading edges at E12.5 and muscleless regions in the dorsal and ventral diaphragm at E17.5. (**C**) Early deletion of fibroblast-derived HGF (tamoxifen at E9.5) at E14.5 results in large muscleless regions, including dorsal and ventral muscle regions, while later mutations (tamoxifen at E11.5) lead to ventral muscleless regions. CM, crural muscle; CT, central tendon; So, somite.

and expansion of diaphragm muscle progenitors. Our experiments conditionally deleting *Hgf* also indicate that migration of diaphragm and shoulder (spinodeltoid and acromiodeltoid) muscle progenitors is *Hgf*-dependent and occurs contemporaneously. Recruitment of shoulder muscle progenitors to the nascent diaphragm has been proposed as important for the evolutionary origin of the mammalian diaphragm (*Hirasawa and Kuratani, 2013*). Thus, our experiments suggest that the evolutionary acquisition of *Hgf* expression in the PPFs may have been a key event that allowed a subset of shoulder muscle progenitors to be recruited to the nascent diaphragm. Our experiments also indicate that continued *Hgf* expression is critical for the full muscularization of the diaphragm. Therefore, once *Hgf* was expressed in the PPFs there may have been selection for its continued expression at the PPF's leading edges to enable expansion of the muscle and complete separation of the thoracic and abdominal cavities. Localization of HGF at the leading edges provides a mechanism to regulate the directional expansion and shape of muscle by promoting survival and motility of muscle progenitors at the dorsal and ventral edges of the muscle. Interestingly, our data indicate that the evolutionary

acquisition of *Hgf* expression in the PPFs is not sufficient to recruit motor neurons to the PPFs and so some other signal(s) must be involved in the evolutionary recruitment of the phrenic nerves. Also, still unknown are the developmental and evolutionary mechanisms driving formation and morphogenetic expansion of the PPFs.

Finally, our study elucidates some of the cellular mechanisms underlying the etiology of CDH. CDH is characterized by defects in the muscularization of the diaphragm, and two sites where muscle is commonly absent are the dorsal-most region of the diaphragm (designated posterior in humans and hernias in this area are called Bochdalek hernias) and the ventral-most diaphragm (anterior in humans and hernias here are called Morgagni hernias) (*Ackerman et al., 2012*; *Irish et al., 1996*; *Kardon et al., 2017*). Our analysis of diaphragms in which HGF/MET signaling is perturbed has found that these two regions are the most likely to have muscularization defects and suggests mechanistically why the dorsal-most and ventral-most diaphragm are most susceptible to muscularization defects. Mutations or variants in any gene or signaling pathway that, similar to HGF/MET, regulates the proliferation, survival, or motility of muscle progenitors will lead to a depletion of the pool of muscle progenitors and the consequent loss of the dorsal and ventral-most diaphragm muscle since these regions develop last (*Figure 8*). Most surprisingly, our analysis revealed that the muscleless connective tissue regions in mice with deletion of *Hgf* in the PPFs or pharmacological inhibition of MET signaling do not herniate. We previously conducted a detailed analysis of mice in which the transcription factor *Gata4* was deleted in the PPFs (*Merrell et al., 2015*). In these mice, muscleless connective tissue regions develop, but these regions always herniate and give rise to herniated tissue covered by a connective tissue sac. Based on our study of *Gata4*, we had proposed that such 'sac' hernias (*Pober, 2007*) develop when localized regions of amuscular tissue develop in juxtaposition with muscularized tissue; the biomechanical difference in strength between these regions allows the abdominal tissues to herniate through the weaker amuscular regions. However, the lack of herniation in the *Hgf* mutants demonstrates that the formation of amuscular connective tissue regions is not sufficient to cause 'sac' hernias. While the formation of amuscular regions is likely a critical step in the formation of 'sac' hernias, other defects in connective tissue integrity are likely necessary to cause these susceptible amuscular regions to actually herniate. Thus, herniation may be a multistep process involving loss of muscle followed by defects in connective tissue strength or elasticity that allow the liver to herniate into the thoracic cavity. Comparison of amuscular connective tissue that maintains its structural integrity to herniated connective tissue may provide further insight into these processes and reveal therapeutic targets in the future.

# Materials and methods

**Key resources table**

| Reagent type (species) or resource | Designation | Source or reference | Identifiers | Additional information |
|---|---|---|---|---|
| Genetic reagent (*Mus musculus*) | Pdgfra^CreERT2^ | *Chung et al., 2018* | RRID:IMSR_JAX:032770 | Dr. Brigid Hogan (Duke University Medical School) |
| Genetic reagent (*M. musculus*) | Prrx1Cre^Tg^ | *Logan et al., 2002* | RRID:IMSR_JAX:005584 | Dr. Clifford Tabin (Harvard Medical School) |
| Genetic reagent (*M. musculus*) | Pax3^Cre^ | *Engleka et al., 2005* | RRID:IMSR_JAX:005549 | Dr. Kurt Engleka (University of Pennsylvania) |
| Genetic reagent (*M. musculus*) | Pax7^iCre^ | *Keller et al., 2004* | RRID:IMSR_JAX:010530 | Dr. Mario Capecchi (University of Utah) |
| Genetic reagent (*M. musculus*) | Pax7^CreER^ | *Murphy et al., 2011* | RRID:IMSR_JAX:017763 | Dr. Gabrielle Kardon (University of Utah) |
| Genetic reagent (*M. musculus*) | Olig2^Cre^ | *Zawadzka et al., 2010* | RRID:IMSR_JAX:025567 | Dr. William Richardson (University College London) |
| Genetic reagent (*M. musculus*) | Hprt-cre | *Tang et al., 2002* | RRID:IMSR_JAX:004302 | Dr. Jeffrey Mann (Monash University) |
| Genetic reagent (*M. musculus*) | Rosa26^LacZ^ | *Soriano, 1999* | RRID:IMSR_JAX:003309 | Dr. Philippe Soriano (Mount Sinai School of Medicine) |

*Continued on next page*

*Continued*

| Reagent type (species) or resource | Designation | Source or reference | Identifiers | Additional information |
|---|---|---|---|---|
| Genetic reagent (*M. musculus*) | *Rosa26^{nTnG}* | *Prigge et al., 2013* | RRID:IMSR_JAX:023537 | Dr. Edward Schmidt (Montana State University) |
| Genetic reagent (*M. musculus*) | *Rosa26^{mTmG}* | *Muzumdar et al., 2007* | RRID:IMSR_JAX:007576 | Dr. Liqun Luo (Stanford University) |
| Genetic reagent (*M. musculus*) | *Rosa26^{Pham/+}* | *Pham et al., 2012* | RRID:IMSR_JAX:018385 | Dr. David Chan (California Institute of Technology) |
| Genetic reagent (*M. musculus*) | *Hgf^{fl}* | *Phaneuf et al., 2004* | MGI:3574633 | Dr. James Wilson (University of Pennsylvania) |
| Genetic reagent (*M. musculus*) | *Met^{fl}* | *Huh et al., 2004* | RRID:IMSR_JAX:016974 | Dr. Snorri Thorgeirsson (National Institutes of Health) |
| Genetic reagent (*M. musculus*) | *Hgf^{Δ/+}* | This paper | | Generated by *HGF^{fl}* crossed to *Hprt-cre* |
| Genetic reagent (*M. musculus*) | *Met^{Δ/+}* | This paper | | Generated by *Met^{fl}* crossed to *Hprt-cre* |
| Antibody | Anti-PAX7 (mouse monoclonal) | DSHB | Cat# PAX7 | Working concentration: 2.4 µg/ml |
| Antibody | Anti-MYOD (mouse monoclonal) | Thermo Fisher | Cat# MA5-12902 | Working concentration: 4 µg/ml |
| Antibody | Anti-MYOSIN (skeletal, fast) MY-32 (mouse monoclonal) | Sigma | Cat# M4276 | Working concentration: 10 µg/ml |
| Antibody | Anti-GFP (chick polyclonal) | Aves Labs | Cat# 2837 | Working concentration: 20 µg/ml |
| Antibody | Anti-cleaved CASPASE-3, Asp175 (rabbit polyclonal) | Cell Signaling | Cat# 9661 | Working concentration: 20 µg/ml |
| Antibody | Anti-NEUROFILAMENT-L (rabbit monoclonal) | Cell Signaling | Cat# 2837 | Working concentration: 0.48 µg/ml |
| Antibody | Anti-HGFR/c-MET (goat polyclonal) | R&D Systems | Cat# AF527 | Working concentration: 10 µg/ml |
| Antibody | Anti-PHOSPHO-MET, PE Conjugate (rabbit monoclonal) | Cell Signaling | Cat# 12468 | Working concentration: 0.5 µg/ml |
| Other | TaqMan *Gata4* | Thermo Fisher | Mm00484689_m1 | Oligonucleo-tides for qPCR; cDNA product size (bp): 84 |
| Other | TaqMan *Myod1* | Thermo Fisher | Mm00440387_m1 | Oligonucleo-tides for qPCR; cDNA product size (bp): 86 |
| Other | TaqMan *Met* | Thermo Fisher | Mm01156972_m1 | Oligonucleo-tides for qPCR; cDNA product size (bp): 74 |
| Other | TaqMan *Hgf* | Thermo Fisher | Mm01135193_m1 | Oligonucleo-tides for qPCR; cDNA product size (bp): 68 |
| Other | TaqMan *Pax7* | Thermo Fisher | Mm01354484_m1 | Oligonucleo-tides for qPCR; cDNA product size (bp): 68 |
| Other | TaqMan *18S rRNA* | Thermo Fisher | 4333760T | Oligonucleo-tides for qPCR; cDNA product size (bp): 187 |
| Other | TaqMan *Fas* | Thermo Fisher | Mm01204974_m1 | Oligonucleo-tides for qPCR; cDNA product size (bp): 76 |
| Other | TaqMan *Trp53* | Thermo Fisher | Mm01731287_m1 | Oligonucleo-tides for qPCR; cDNA product size (bp): 133 |
| Other | TaqMan *Map1lc3a* | Thermo Fisher | Mm00458724_m1 | Oligonucleo-tides for qPCR; cDNA product size (bp): 63 |

| Reagent type (species) or resource | Designation | Source or reference | Identifiers | Additional information |
|---|---|---|---|---|
| Chemical compound, drug | BMS777607 | Selleckchem | Cat# S1561 | In vivo: 0.05 mg/g of body weight In vitro: 10 μM |

## Mice and staging

All mouse lines have been previously published. We used $Prrx1Cre^{Tg}$ (**Logan et al., 2002**), $Pdgfra^{CreERT2}$ (**Chung et al., 2018**), $Pax3^{Cre}$ (**Engleka et al., 2005**), $Pax7^{iCre}$ (**Keller et al., 2004**), $Pax7^{CreER}$ (**Murphy et al., 2011**), $Olig2^{Cre}$ (**Zawadzka et al., 2010**), and $Hprt$-$cre$ (**Tang et al., 2002**) Cre alleles. Cre-responsive reporter alleles included $Rosa26^{LacZ}$ (**Soriano, 1999**), $Rosa26^{nTnG}$ (**Prigge et al., 2013**), $Rosa26^{mTmG}$ (**Muzumdar et al., 2007**), and $Rosa26^{Pham/+}$ (**Pham et al., 2012**). The $Hgf^{fl}$ (**Phaneuf et al., 2004**) conditional allele (B6;129-Hgftm1Jmw/Mmnc, identification number 423-UNC) was obtained from the Mutant Mouse Regional Resource Center, an NIH-funded strain repository, and was donated to the MMRRC by S. E. Raper, Ph.D., University of Pennsylvania Medical Center. We also used the $Met^{fl}$ (**Huh et al., 2004**) conditional allele. $Hgf^{\Delta/+}$ and $Met^{\Delta/+}$ mice were generated by breeding $Hgf^{fl}$ and $Met^{fl}$ mice to $Hprt$-$cre$ mice. Embryos were staged as E0.5 at noon on the day dams presented with a vaginal plug. Mice were backcrossed onto a C57Bl/6J background. Experiments were performed in accordance with protocols approved by the Institutional Animal Care and Use Committee at the University of Utah.

## Immunohistochemistry, immunofluorescence, and in situ hybridization

For section immunofluorescence, optimal cutting temperature (OCT)-embedded tissues were sectioned to 10 μm thickness and fixed for 5 min in 4% paraformaldehyde (PFA). Tissue sections were blocked for 60 min in 5% goat serum in phosphate-buffered saline (PBS), incubated overnight at 4°C in primary antibodies. Sections were washed in PBS, incubated with secondary fluorescent antibodies (used at 1–5 μg/ml; Jackson Laboratories or Thermo Fisher) for 2 hr at room temperature (RT), washed with PBS, stained for 5 min with Hoechst to label nuclei, post-fixed in 4% PFA, rinsed in water and mounted with Fluoromount-G (Southern Biotech). Primary antibodies are listed in Key resources table. Sections were imaged on an Olympus BX63.

For immunofluorescence on PPF cell cultures, cells were fixed in 4% PFA for 20 min at RT, washed in PBS, blocked for 60 min in 5% goat serum with 0.1% Triton X-100 in PBS, and stained overnight for primary antibodies (see Key resources table). Cells were then washed in PBS, incubated for 2 hr in secondary antibodies, washed in PBS, incubated in Hoechst to label nuclei, washed in PBS, and rinsed in water and mounted in Fluoromount. EdU (Life Technologies) was applied to cells 1 hr prior to fixation and detected after secondary labeling based on the manufacturer's instructions with Alexa647 picolyl azide. Stained cells were imaged with ImageXPress Pico automated cell imager (Molecular Devices).

Whole-mount embryos were fixed for 24 hr in 4% PFA at 4°C, dissected, incubated for either 2 hr at RT or overnight at 4°C in Dent's bleach (1:2 30% $H_2O_2$:Dent's fix) and stored in Dent's fix (1:4 DMSO:methanol) for at least 5 days at 4°C. Embryos were washed in PBS, blocked for 1 hr in 5% goat serum and 20% DMSO, incubated in primary antibodies (see Key resources table) for 48 hr, washed in PBS, incubated in secondaries for 24–48 hr, washed in PBS, and cleared BA:BB (33% benzyl alcohol, 66% benzyl benzoate) at RT. Embryos labeled with AP-conjugated anti-Myosin heavy chain were heat-inactivated at 65°C for 1 hr, incubated in primary antibody for 48 hr, and detected with 250 μg/ml NBT and 125 μg/ml BCIP (Sigma) in alkaline phosphatase buffer. For detection of HRP-conjugated secondary antibodies, embryos were incubated in 10 mg diaminobenzidine tetrahydrochloride in 50 ml PBS with 7 μl hydrogen peroxide for approximately 20 min.

For whole-mount EdU analysis in embryos, 10 μg/g of body weight of EdU was administered to pregnant females 1 hr prior to harvest via IP injection.

Whole-mount in situ hybridization was performed as previously described (**Riddle et al., 1993**). For whole-mount β-galactosidase staining, embryos were fixed overnight in 1% PFA at 4°C and 2 mM $MgCl_2$. Diaphragms were dissected, washed in PBS and then in LacZ rinse buffer (100 mM sodium phosphate, 2 mM $MgCl_2$, 0.01% sodium deoxycholate, and 0.02% Ipegal), and stained for 16 hr at 37°C in X-gal staining solution (5 mM potassium ferricyanide, 5 mM potassium ferrocyanide, and 1 mg/ml X-gal).

## Microscopy and three-dimensional rendering

Whole-mount fluorescent images were taken on a Leica SP8 confocal microscope. Optical stacks of whole-mount images were rendered and structures highlighted using FluoRender (*Wan et al., 2009*). To highlight features (such as the phrenic nerve in *Figure 4*), objects were selected in FluoRender based on morphology on individual Z optical sections using the paintbrush tool, and then these objects were extracted, rendered, and pseudo-colored. For EdU and cleaved Caspase-3 labeling, PPFs were first selected based on morphology and the total PPF area measured. Individual nuclei from PPFs were then counted using the Component Analyzer Tool.

## Cell culture, media, and reagents

E12.5 embryos were dissected from pregnant $Rosa26^{mTmG/mTmG}$ or $Rosa26^{nTnG/nTnG}$ females mated with $Pax3^{Cre/+}$ males. Embryos and PPFs were dissected as previously described (*Bogenschutz et al., 2020*). Briefly, embryos were dissected from yolk sacs in DMEM/F-12 GlutaMAX (Invitrogen) pre-warmed to 37°C. To isolate the trunk region, embryos were cut posterior to the forelimbs and just anterior to the hindlimbs, leaving liver largely attached to the trunk. Heart and lungs were removed from the thoracic cavity with forceps and the trunk trimmed to expose the PPFs sitting cranial to the liver. Trunks were pinned to a 6 mm dish coated in Sylgard and PPF pairs manually isolated with forceps from the body wall, septum transversum, and underlying liver. Each PPF explant pair was then placed in a single well from a 96-well plate with 100 µl of media. Growth of both PPF fibroblasts and myogenic cells was promoted using DMEM/F-12 GlutaMAX (Invitrogen), 10% FBS, 50 µg/ml gentamicin, and 0.5 nM FGF. PPFs were grown in a 37°C incubator overnight and then imaged on an ImageXPress Pico automated cell imager (Molecular Devices) or a Leica SP8 confocal microscope, for 1–4 days, changing media in the wells every 2 days.

## Chemical treatments

For BMS777607 administration to pregnant females, 0.05 mg/g of body weight (e.g., 1 mg BMS777607 for a 20 g mouse) in 70% PEG-300 in PBS was administered via oral gavage. Vehicle alone (1% DMSO in 70% PEG300 in PBS) was administered to control pregnant dams. For cell culture experiments, BMS777607 was used at 10 µM concentration in 0.001% DMSO, and vehicle controls were treated with 0.001% DMSO.

## Cell growth, motility, and shape analysis

Proliferating myogenic cells were imaged using the ImageXPress Pico that took GFP, Tomato, and phase images every 3 hr of the entire PPF sample for 72 hr total. CellReporterXpress (Molecular Devices) software was then used to count GFP+ cells per time point to calculate growth of myogenic cells over time. To control for differences in initial number of myogenic cells per well, fold changes of cell growth were calculated by dividing each treatment by the initial cell number at time 0. For cell motility analysis of $Pax3^{Cre/+}$; $Rosa26^{nTnG/+}$ nuclei, GFP and phase images were taken every 8 min for 14 hr. Tracking, cell velocity, and displacement of peripheral cells (as an analog for the leading edge of the PPFs) were determined using TrackMate (*Tinevez et al., 2017*). For cell shape analysis on $Pax3^{Cre/+}$; $Rosa26^{mTmG/+}$ membranes, GFP and phase images were imaged every 4 min apart for 68 min total. Circularity of peripheral cells was analyzed in Fiji.

## Tamoxifen injections and muscle injury

$Pax7^{CreERt2/+}Rosa26^{Pham/+}$ mice (*Pham et al., 2012*) were given five 2 mg doses (10 mg total) of tamoxifen (Cayman Chemical, 13258) (TAM) by intraperitoneal injection prior to injury. Barium chloride (25 µl 1.2% in sterile demineralized water) was injected into the tibialis anterior muscle of each mouse with a Hamilton syringe similar to *Murphy et al., 2014*.

## FACS cell isolation and sorting

Isolation of mononuclear myogenic cells from adult tibialis anterior muscle was performed as described previously (*Murphy et al., 2014*). Tibialis anterior muscles were dissected, minced, and digested for 1 hr at 37°C in 100 µl of 5 mg/ml liberase (Sigma-Aldrich, 5401127001) and 25 µl of 10 U/µl DNAseI (Sigma-Aldrich, 4716728001) in 3 ml Ham's F12 media (Thermo Fisher Scientific, 11765054). Samples were passed through 70 µm and 40 µm filters, centrifuged at 1800 rpm for 10 min, aspirated

supernatant, and pellet resuspended in satellite cell growth media (15% horse serum [Gibco, 16050-122], 1:1000 50 mg/ml gentamicin [Thermo Fisher Scientific, 15750060] in F12 media). Myogenic mononuclear cells were isolated and sorted via GFP on Propel Labs Avalon (Bio-Rad).

## Phospho flow cytometry

Sorted GFP+ cells were washed with Ham's F12 media, centrifuged at 800 × $g$ for 5 min at 4°C and aspirated supernatant. Cells were fixed with 1.5% PFA for 15 min at RT, washed with PBS, centrifuged at 800 × $g$ for 5 min at 4°C, and supernatant was aspirated. The cell pellet was resuspended in 100% methanol cooled to –20°C, vortexed for 30 s, and incubated on ice for 30 min. Cells were washed with 0.5% BSA in PBS with sodium azide for 5 min, centrifuged at 800 × $g$ for 5 min at 4°C, aspirated supernatant. Cells were washed with 0.5% BSA with sodium azide for 5 min, centrifuged at 800 × $g$ for 5 min at 4°C, aspirated supernatant, and resuspended into 0.5% BSA with sodium azide. pMET conjugated to phycoerythrin (PE) (Cell Signaling, 12468) primary antibody 1:50 dilution was added to the samples for 20 min at RT. Samples were washed with PBS, centrifuged at 800 × $g$ for 5 min at 4°C, aspirated supernatant, and resuspended in PBS for analysis on BD FACSCanto II HTS (BD Biosciences). Median PE frequency from FloJo was used for statistical analysis.

## RNA extraction, cDNA synthesis, and quantitative polymerase chain reaction (qPCR)

The *Quick*-RNA Microprep Kit (Zymo, Irvine, CA) was used to extract total RNA according to the manufacturer's protocol. Applied Biosystems High-Capacity RNA-to-cDNA kit (Thermo Fisher) was used to synthesize cDNA from purified RNA according to the manufacturer's protocol. qRT-PCR was used to analyze expression of *Met, Pax7, Hgf, MyoD1,* and *Gata4* using pre-validated primer sets (TaqMan, Thermo Fisher; Key resources table). 10 µl reaction volumes were prepared using TaqMan Fast Advanced Master Mix (Thermo Fisher). The following conditions were used for amplification: 20 s at 95°C followed by 40 cycles at 95°C for 1 s, and 60°C for 20 s. Gene expression levels were normalized against *18S* ribosomal RNA for each sample and fold changes calculated using $2^{-\Delta/\Delta Ct}$ method (*Schmittgen and Livak, 2008*) by setting expression levels of each gene in DMSO-treated cell culture as 1. Data from three biological replicates were calculated and plotted as average fold changes with standard error of the mean (SEM).

## Statistical analysis

Data are presented as ± SEM. For growth comparison between chemical treatments, repeated ANOVA analysis was run on the log2 fold change of GFP+ cells to normalize the distribution of cell growth over time. Unpaired two-tailed *t*-tests or one-way ANOVA were used for other statistical analyses.

## Acknowledgements

We thank N Burns for critical reading of the manuscript and Y Wan for help with FluoRender analysis. We are grateful to P Pandey for technical assistance. qPCR experiments were performed by the University of Utah Genomics Core Facility and confocal imaging was performed at the University of Utah Cell Imaging Core with assistance of Xiang Wang. This work was supported by the National Institutes of Health R01HD087360 to GK, F32 HD093425 and 1K99HD101682 to EMS and Wheeler Foundation to GK.

## Additional information

### Funding

| Funder | Grant reference number | Author |
| --- | --- | --- |
| National Institutes of Health | R01HD087360 | Gabrielle Kardon |
| National Institutes of Health | F32 HD093425 | Elizabeth M Sefton |

| Funder | Grant reference number | Author |
|---|---|---|
| National Institutes of Health | K99HD101682 | Elizabeth M Sefton |
| National Institutes of Health | R01 HD104317 | Gabrielle Kardon |
| Wheeler Foundation | | Gabrielle Kardon |

The funders had no role in study design, data collection and interpretation, or the decision to submit the work for publication.

## Author contributions

Elizabeth M Sefton, Conceptualization, Formal analysis, Funding acquisition, Investigation, Visualization, Methodology, Writing - original draft, Project administration, Writing - review and editing; Mirialys Gallardo, Formal analysis, Investigation, Visualization, Methodology; Claire E Tobin, Mary P Colasanto, Investigation; Brittany C Collins, Investigation, Writing - review and editing; Allyson J Merrell, Resources (Figure 3 – figure supplement 2 Panels G and H); Gabrielle Kardon, Conceptualization, Resources, Data curation, Formal analysis, Supervision, Writing - original draft, Project administration, Writing - review and editing

## Author ORCIDs

Elizabeth M Sefton ⓘ http://orcid.org/0000-0001-6481-612X
Gabrielle Kardon ⓘ http://orcid.org/0000-0003-2144-4463

## Ethics

Experiments were performed in accordance with protocols (#1435) approved by the Institutional Animal Care and Use Committee at the University of Utah.

## Decision letter and Author response

Decision letter https://doi.org/10.7554/eLife.74592.sa1
Author response https://doi.org/10.7554/eLife.74592.sa2

# Additional files

## Supplementary files

• Transparent reporting form

## Data availability

Numerical and source data used to generate figures have been included in the Source Data file for Figures 1-7; Figure 1-figure supplement 1, Figure 3-figure supplement 1, Figure 3-figure supplement 2, Figure 3-figure supplement 1, Figure 4-figure supplement 1, Figure 5-figure supplement 1, and Figure 6-figure supplement 1.

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
