## [Editor Report]

It was previously known that HGF and Met control delamination and migration of muscle progenitor cells that colonize the diaphragm. The article by Sefton and coworkers confirms and extends these observations using conditional mouse lines in which the HGF gene was targeted by Cre/loxP recombination, and Met inhibitors that are applied at different stages of development. Together these new data show that HGF derived from the pleuroperitoneal folds is directly required for the recruitment of Met+ muscle progenitors to the diaphragm and continues to be essential for the survival of a pool of progenitors that eventually form the diaphragm muscle. Moreover, the authors show the effects of the diaphragm muscle on the development of the phrenic nerve that innervates this muscle, in particular, in the absence of muscle deficits in the branching of the phrenic nerve are observed. Overall, the technical quality of the data on diaphragm muscle development and its effect on the branching of the phrenic nerve are excellent.

---

## [Decision Letter]

**Decision letter after peer review:**

Thank you for submitting your article "Fibroblast-derived HGF integrates muscle and nerve development during morphogenesis of the mammalian diaphragm" for consideration by *eLife*. Your article has been reviewed by 3 peer reviewers, including Carmen Birchmeier-Kohler as Reviewing Editor and Reviewer #1, and the evaluation has been overseen by Didier Stainier as the Senior Editor. The following individual involved in review of your submission has agreed to reveal their identity: Thomas Braun (Reviewer #2).

Summary

Sefton et al., analyze 'Fibroblast-derived HGF integrates muscle and nerve development during morphogenesis of the mammalian diaphragm'. The role of Met in the development of the muscle has been previously investigated by many papers, and roles in delamination of progenitor cells form the somites, in migration, survival, proliferation and differentiation have been described. The novel finding of the paper is a in depth analysis of the development of the diaphragm muscle, and the role of Met and HGF in the process. The authors provide evidence for the expression of HGF in pleuroperitoneal folds and for its requirement for muscle progenitor expansion and maintenance during diaphragm muscle formation. The reviewers found the manuscript clearly written and the experimental approach overall of high standard. However, there were several deficiencies noted that need to be addressed. The data supporting that HGF/Met function during development of the phrenic nerve was found to be less well supported and requiring additional experimental data to be included in the manuscript.

Essential revision

1) A major concern is the limited data on the role of Met in the development of the phrenic nerve. While it is well documented that HGF acts as a trophic factor for motor neurons in culture, its role in development of motor neurons has been highly debated. Moreover, careful genetic analyses previously demonstrated indirect mechanisms of Met during motor neuron development. Despite the weakness of the data on the role of Met in phrenic development, the role of HGF/Met in is strongly emphasized in abstract /intro/discussion. Data relying on motor neurons specific ablation of met need to be included in order to strengthen this point. Additional suggestions how to improve the analysis of phrenic development in Met-/+ animals are provided in the individual reviews appended below.

2) The authors should exclude a connective tissue phenotype in conditional HGF/Met mutants. Why is Prx1Cre used instead of PDGFRaCre to trace PPFs? In addition, the reason for the differences in phenotypes regarding dorsal/ventral diaphragm muscles after either HGF inactivation or Met-inhibitor treatment should be explained, or at least discussed. Is this due to timing of the mutation/inhibition, or to efficacy of ablation/inhibition?

3) The authors propose a role of Met in myogenic commitment, based on the co-culture experiments. Myogenic commitment implies a major role in myogenic lineage progression or differentiation. The evidence for that claim is weak and the reduced differentiation might be the consequence of increased cell death, reduced proliferation or other causes that change myoblast density. It is mandatory to address this issue appropriately, by analyzing both, fibroblast and myoblast cell numbers, proliferation, apoptosis, myogenic differentiation (i.e. ratio of MyoD/MyoG-positive cells).

4) A potentially interesting conclusion form the manuscript is hat hernia does only develop when both, connective tissue and muscle of the diaphragm are affected. Although this explanation is intriguing, the authors have to make sure that body wall muscles are fully developed, to exclude that the abdominal pressure is not reduced in mice conditional Met/HGF mutations. The images in Figures 1 and 2 seem to indicate normal development of body wall muscles but a dedicated statement in this respect would be helpful.

5) Treatment of WT embryos from E7.5 will impact on myogenic progenitor delamination, and this is also expected for Pax3CreKIMetnull/fl mutation. The authors have to dissociate the Met-related somite delamination, a migration of the progenitors to the diaphragm and the potential additional role of Met signaling in the colonization and expansion of myogenic progenitor cells in the anlage of the diaphragm.

6) Validation for the drug BMS777607 effectiveness in inhibiting MET downstream pathway should be performed and shown.

7) Comments referring to missing controls, scale bars, statistics need to be addressed.

Additional points: The authors should address wherever possible the other comments raised by the three reviewers (see below).

*Reviewer #1 (Recommendations for the authors):*

It was previously shown that HGF and Met controls development of the diaphragm muscle. In particular, the signal induces delamination and migration of muscle progenitor cells that colonize the diaphragm. The present manuscript by Sefton and coworkers confirms and extends these observations using (i) conditional mouse lines in which the HGF gene was targeted by Cre/loxP recombination in the pleuroperitoneal folds (Prx1-cre) and at other sites PdgfraCreERT2, and of (ii) Met inhibitors. Overall, the technical quality of the data on diaphragm muscle development is excellent; the conceptual advance over previous work is not exceptional; the evidence for Met/HGF-dependent development of the phrenic nerve is marginal and needs to be strengthened.

The data show that fibroblasts provide HGF signals received by Met in muscle progenitor cells that is essential for diaphragm development. The PdgfraCreERT2 line was used to demonstrate that HGF produced by fibroblasts but not by muscle progenitors is essential for diaphragm development. Moreover, development of dorsal and ventral regions of diaphragm muscle requires continuous MET signaling. Thus, HGF is not only required for the delamination of progenitors, but also for proliferation and survival of those muscle progenitors that reached the anlage of the diaphragm.

My major concern is the limited data on the HGF-dependent development of the phrenic nerve (defasciculation). While it is well documented that HGF acts as a trophic factor for motor neurons in culture, its role in development of motor neurons was highly debated due to the fact that some changes observed in Met or HGF mutant mice in vivo are also present in other mutants that lack the muscle groups derived from migrating muscle progenitors. Moreover, careful genetic analyses previously demonstrated indirect mechanisms of Met during motor neuron development, i.e. a non-cell-autonomous function of Met during the recruitment of motor neurons to PEA3-positive motor pools (Helmbacher et al., Neuron 2003).

Sefton et al., provide an analysis of a single time point, one histological picture (3G, magnified in 3H) that indicate that in Met+/- animals defasciculation of the phrenic nerve does not occur correctly. This is accompanied by a quantification that barely reaches significance (Figure 3K). Data shown in Figure 7 using Met inhibitors show a major change in phrenic nerve branching, which is presumably due to the major change in diaphragm development, as conceded by the authors.

Despite this weakness on the experimental side, the role of HGF/Met in phrenic nerve development is strongly emphasized in abstract /intro/discussion (e.g. line 414: However, PPF-derived HGF is crucial for the defasciculation and primary branching of the nerve, independent of muscle). The data need to be strengthened in order to conclude that HGF coordinates both, diaphragm muscle and phrenic development. I expect that the defasciculation of the phrenic nerve is highly dependent on the developmental stage. The authors should provide data that show different stages of phrenic nerve development, i.e. the time course of the of defasciculation in wildtype animals, explain how the staging was done, and compare different stages in the Met mutants and analyze whether the defasciculation is resolved at later stages. Met mutations also affect placental development, resulting in developmental delays that in turn might lead to an apparent small change in the time course of defasciculation. The authors should exclude that indirect effects cause the small change in phrenic nerve morphology, for instance by examining conditional Met mutations that are restricted to motor neurons. Good Cre lines that target motor neurons are available.

*Reviewer #2 (Recommendations for the authors):*

Since the authors observed a correlation between loss of should muscle and loss of diaphragm muscularization, which is related to the timing of migration of shoulder muscle and diaphragm muscle progenitors, they claim a "closer relationship" between should muscles and diaphragm, which was further extended in the discussion. I was not convinced by this conclusion. Is there a "closer relationship" between the muscles just because the progenitor cells migrate roughly at the same time? They authors may modify or delete this statement, although I agree that a broader expression of HGF may facilitate enhanced recruitment of muscle progenitor cells, required for formation of the diaphragm.

Based on co-cultures of PPFs and myoblasts the authors describe a function of MET in myogenic commitment. In my opinion the evidence for such a function is weak. The authors observed no change in proliferation but reduced motility and a higher rate of apoptosis after pharmacological inhibition of MET. Increased apoptosis and reduced aggregation of myoblasts at distinct locations may easily interfere with myogenic differentiation. A general function of MET in myogenic commitment does not seem very likely, since myoblasts that do not undergo prior long-range migration differentiate normally in the absence of Met.

Pharmacological inhibition of MET increases the rate of apoptosis in numerous cell types, which has been studied extensively in cancer. It seems appropriate to explore the mechanism of increased apoptosis in myoblasts following MET inhibition more closely. Previous reports suggest increased expression of p53, increased sensitivity to Fas-mediated apoptosis or increased autophagy, among others, as potential causes for increased apoptosis after MET inhibition. Which pathway is relevant in myoblasts?

Obviously, it would be great to learn more about the mechanisms that control Hgf expression in fibroblasts within and derived from PPTs. Such knowledge may also help to better understand the reasons leading to Congenital Diaphragmatic Hernias (CDH). Unfortunately, the authors did not to go any further in this direction.

The authors observed that branching of the phrenic nerve was reduced in heterozygous Met mutants with normal diaphragm musculature, suggesting a direct role of Met in phrenic nerve branching. Timed inactivation of Met specifically in motoneurons would greatly increase the impact of this finding and allow a more specific analysis of the role of MET in phrenic nerve development and branching.

Surprisingly, the authors did not observe CDH in mutant mice with muscle-less diaphragms, from which they conclude that additional defects in the connective tissue are necessary to allow hernia formation. Although this explanation is intriguing, the authors have to make sure that abdominal pressure is not reduced in mice without muscularized diaphragms, e.g. demonstrate that body wall muscles are fully functional. The images in Figures 1 and 2 seem to indicate normal development of body wall muscles but a dedicated statement in this respect would be helpful.

Scale bars are missing in some panels.

According to the methods part, Student's t-test was used for the statistical analysis shown in Figure 3. A pairwise comparison of WT, heterozygous and homozygous Met mutants is not appropriate when all three genotypes are compared with each other. An ANOVA test should be used as in Figure 5F.

*Reviewer #3 (Recommendations for the authors):*

The present manuscript addresses questions on the role of HGF/MET signaling in diaphragm formation once myogenic progenitors have already migrated to the PPFs. In addition, it identifies the PPFs as the source of HGF. The study is interesting and developed with rigor. However, the role of HGF is not clearly dissociated from the presence/absence of the fibroblasts/connective tissue itself. Also, the authors do not link in the results, for example, how the muscleless diaphragms and HGF itself relate to the hernia phenotype mentioned in the abstract.

Figure 1

1C) Co-staining for Hgf (PPFs) and Met (migrating progenitors) should be provided for a clear visualization of ligand-expressing cells and receptor-expressing cells.

1F) Co-staining for Met and migrating progenitors (PAX3 or LBX1 for example) should be provided for a clear visualization of migrating progenitors versus general Met expression in other cell types.

1M) The claim that fibroblast-derived HGF is required for diaphragm muscle development cannot be addressed with only this experimental analysis. What is the connective tissue phenotype in PDGFaCreER;HGFnull/fl embryos? Lack of connective tissue could affect muscle migration and development rather than HGF expression on its own. Are fibroblasts still present?

1G, I, K, M) Control genotypes should be analyzed and added in the Figure or as Supplementary Data.

1I) Pax3CreKIMetnull/fl originates a muscleless diaphragm but this could be associated with lack of delamination and migration from the somites rather than a specific MET requirement for diaphragm muscle formation once progenitors have colonized this area as suggested by the authors.

Figure 3

It is not clear whether HGF controls phrenic nerve formation independently of muscle. In the end of this section the authors mention that Met heterozygous embryos have normal muscles referring to Figure 1K (which is not a picture referring to a Met het embryo). The authors should confirm if in Met null embryos there is a direct effect on the nerve bifurcation and brunching or if the lack of muscle is leading to this observation. Is the phenotype dose dependent for muscle formation? This is not properly shown by the authors. Could the authors perform a conditional KO of Met in the nerves?

This concern is further supported by the Figure 4 data where phrenic nerve only extends to regions with muscle.

Figure 4

The role of HGF versus the presence of fibroblasts/connective tissue remains vague. Is the lack of ventral muscles associated with lack of migration of progenitors within the forming diaphragm towards an HGF source? Or due to the lack of connective tissue scaffold itself? Why is Prx1Cre used instead of PDGFRaCre (the one used in the actual experiments) to trace PPFs?

4E) Please provide a control with a Cre allele to compare putative secondary effects for the presence of the Cre in the cKO embryos.

Figure 5

Validation for the drug BMS777607 effectiveness in inhibiting MET downstream pathway should be performed and shown.

The phenotype with the inhibition of Met signaling (lack of dorsal muscles in addition to ventral) is distinct to the one observed in Figure 4 (less ventral muscle formation in PDGFaCreER;HGFnull/fl). How do the authors explain the dorsal phenotype when inhibiting Met (since migration should have not been affected at this time-point) and Hgf inhibition in Figure 4 is not leading to this phenotype?

Figure 6

What is the time point after treatment in B and C? What is the total nuclei number in the cultures (since these are co-cultures)?

The authors should provide pictures together with nuclear staining for a global view of the cell density in the cultures. Also T0 time point pictures should be added for comparison, in particular in C.

In these experiments the authors performed co-culture (fibroblasts + myogenic progenitors) derived from PPfs. However, fibroblast phenotype is not addressed. If there is less GFP+ cells is there more fibroblasts in the culture? Could this impact on the phenotype observed and linked by the authors to the Met inhibitor treatment? The only piece of evidence is Gata4 qPCR which is not sufficient to address fibroblast phenotype in the culture.

Figure 7

Treatment of WT embryos from E7.5 will impact on myogenic progenitor migration. The authors have to dissociate the Met-related somite migration phenotype and the potential additional role of Met signaling in the diaphragm muscle formation.

[Editors’ note: further revisions were suggested prior to acceptance, as described below.]

Thank you for resubmitting your work entitled "Fibroblast-derived *HGF* controls recruitment and expansion of muscle during morphogenesis of the mammalian diaphragm" for further consideration by *eLife*. Your revised article has been evaluated by Didier Stainier (Senior Editor) and a Reviewing Editor.

The manuscript has been much improved but there are a few remaining issues that need to be addressed, as outlined below:

*Reviewer #1 (Recommendations for the authors):*

The majority of my concerns were appropriately addressed by the authors. There are a few additional points that need to be addressed, and most of these concern the text/wording.

Unfortunately, the changes in the text have not been highlighted in red in the uploaded version of the manuscript, as it was mentioned in the rebuttal letter.

1. While the quality of the figures is overall good, many of the panels in Figure 2 are very dark. It should be easy to modify this.

2. While the authors show very convincingly that the effects on the phrenic nerve observed are caused indirectly by the loss of muscle, and not directly by the loss of HGF/Met signaling, this is not always made clear in the text. Furthermore, the mechanism that causes branching deficits should be clearly stated in the Abstract.

Line 41-42...and indirectly required for phrenic nerve primary branching.

Please mention the specific indirect mechanisms, i.e. via the effect on muscle development.

Line 81…HGF is also critical for innervation.

The reports cited do not distinguish between the effect of the nerve/muscle or report in vitro experiments. The text should take this into account.

238: Thus, PPF-derived HGF is necessary for phrenic nerve defasciculation.

Additional experiments shown in the next paragraph indicate that it is the absence of the muscle that causes defasciculation. Please rephrase.

3) Met inhibition using BMS777607

BMS777607 acts as an AXL, RON and Met tyrosine kinase inhibitor. This should be mentioned. Are effects of RON/AXL on skeletal muscle development described?

4) Antibody specificity: Antibody specificity should be tested by analysis of the phrenic nerve on Met mutants.

*Reviewer #2 (Recommendations for the authors):*

Sefton et al., have submitted a revised version of a study, in which the role of fibroblast-derived HGF for recruitment and expansion of muscle during morphogenesis of the mammalian diaphragm was investigated. The authors have changed the title to cope with new findings, indicating that reduced primary branching of the phrenic nerve in Met-mutants is not due direct effects of HGF on the nerve but most likely caused by indirect effects resulting from reduced muscle formation in Met-mutants.

The authors did an excellent job to deal with the reviewers' criticisms. In particular, by specifically deleting Met in motoneurons and by analysis of splotch mutants they demonstrate that Met does not intrinsically regulate phrenic nerve branching. Analysis of splotch mice, which show normal expression of HGF PPFs but display muscle-less diaphragms, revealed reduced branching of the phrenic nerve, similar to Met-mutants, clearly suggesting a critical role of muscle fibers in the diaphragm for phrenic nerve branching. Furthermore, Sefton et al., now demonstrate that fibroblasts expand and populate the diaphragm in the absence of muscle, which essentially excluded a connective tissue phenotype in conditional Hgf/Met mutants. They also clarified the statement about what I understood was meant to claim a role of Hgf/Met in myogenic commitment. Additional controls (effectiveness of BMS777607 for inhibition of MET, treatment with BMS777607 at additional timepoints) were done as requested, providing additional interesting insights into a role of Met for ventral expansion of the diaphragm at relatively late developmental timepoints.

The authors argue that "PPF-derived HGF, via muscle, controls phrenic nerve defasciculation". Well, this is formally correct but exaggerates the role PPF-derived for phrenic nerve defasciculation in my view. Probably, deletion of any gene that prevents diaphragm muscle formation will have similar effects on phrenic nerve defasciculation. I would prefer a more neutral statement: "Muscle formation, requiring PPF-derived HGF, controls phrenic nerve defasciculation", or something similar. The authors may consider modifying the statement.

I do not have any further objections.

---

## [Author Response]

Essential Revision1) A major concern is the limited data on the role of Met in the development of the phrenic nerve. While it is well documented that HGF acts as a trophic factor for motor neurons in culture, its role in development of motor neurons has been highly debated. Moreover, careful genetic analyses previously demonstrated indirect mechanisms of Met during motor neuron development. Despite the weakness of the data on the role of Met in phrenic development, the role of HGF/Met in is strongly emphasized in abstract /intro/discussion. Data relying on motor neurons specific ablation of met need to be included in order to strengthen this point. Additional suggestions how to improve the analysis of phrenic development in Met-/+ animals are provided in the individual reviews appended below.

In response to comments from the reviewers, we have more thoroughly investigated the role of Met in the development of the phrenic nerve and include two new sets of genetic experiments. In our first submission, we found a decreased number of phrenic nerve branches at E11.5 in *Met ^Δ/ Δ^* and *Met ^Δ/+^* compared with *Met^+/+^* embryos. In the *Met ^Δ/ Δ^* embryos, no muscle is present in the diaphragm. Therefore, the greatly reduced branching in these embryos is likely a secondary effect of the requirement of Met in muscle progenitors for diaphragm muscularization. Of particular interest is the reduced branching in the *Met ^Δ/+^* embryos. Because the diaphragm is muscularized in these embryos, this suggested that Met may be required intrinsically in the phrenic nerve. One reviewer suggested that the reduced branching in the *Met ^Δ/+^* embryos could be due to a developmental delay in the whole embryo. However, we found that *Met ^Δ/ Δ^* and *Met ^Δ/+^* embryos are not overall delayed relative to *Met^+/+^* embryos (as measured by crown rump length or limb length; Figure 3—figure supplement 1). Also, to increase the robustness of these data, we added additional embryos to the analysis. We then extended our analysis of *Met ^Δ/ Δ^*, *Met ^Δ/+^* and *Met^+/+^* embryos to E12.5 (Figure 3—figure supplement 1) to see whether the branching phenotype persisted; we found that while the of *Met ^Δ/ Δ^* embryos continue to have very few branches, the number of branches in *Met ^Δ/+^* embryos recovers and matches that of *Met^+/+^* embryos.

To explicitly test whether Met is required within the phrenic nerve, we used *Olig2^Cre/+^*to conditionally delete *Met*. This line was chosen for its early expression in motor neurons (Zawadzka et al., 2010). We examined *Olig2^Cre/+^;Met*
^Δ*/flox*^ embryos compared to *Olig2^Cre/+^; Met^flox/+^* embryos. We chose to include *Olig2^Cre^* in our controls because the *Olig2^Cre^* is a knock-in/knock-out and *Olig2* has important roles in nerve development. However, deletion of *Met* did not affect the number of branches at E11.5 (Figure 3—figure supplement 2) or E12.5 (data not shown). These data suggest that Met does not intrinsically regulate phrenic nerve branching. This suggests that PPF-derived HGF regulates phrenic nerve branching indirectly via muscle. To test if HGF is sufficient to promote early stages of nerve branching in the absence of muscle, we turned to *Pax3^SpD/SpD^* mutants in which a point mutation in Pax3 prevents migration of muscle progenitors into the diaphragm (Figure 3—figure supplement 2). In these embryos, the diaphragm is muscleless, but the PPFs still express HGF. In these diaphragms the number of branches at E11.5 is severely reduced. These data demonstrate that in the absence of muscle the presence of HGF in the PPF fibroblasts is not sufficient to support diaphragm branching.

Altogether our data demonstrate that PPF-derived HGF, via its regulation of muscle, controls the primary branching of phrenic nerve. The *Met ^Δ/+^* data demonstrate that Met controls phrenic nerve branching at E11.5 in a dose-dependent manner, but this effect is lost by E12.5. Although we see no obvious defects in muscle of *Met ^Δ/+^* diaphragms at later stages, the most parsimonious explanation of the reduced phrenic nerve branching at E11.5 is that this is due to fewer muscle progenitors at this time point.

We thank the reviewers for prompting us to look at the role of HGF/Met in the phrenic nerve more closely. Our revised conclusions are presented in the Results and Discussion. We show that PPF-derived HGF is critical for integrating both muscle and phrenic nerve development, but now demonstrate that HGF’s regulation of phrenic nerve branching is via muscle, which is well-known to express multiple trophic factors required by motor neurons.

2) The authors should exclude a connective tissue phenotype in conditional HGF/Met mutants. Why is Prx1Cre used instead of PDGFRaCre to trace PPFs? In addition, the reason for the differences in phenotypes regarding dorsal/ventral diaphragm muscles after either HGF inactivation or Met-inhibitor treatment should be explained, or at least discussed. Is this due to timing of the mutation/inhibition, or to efficacy of ablation/inhibition?

*Prx1Cre* and *PDGFRa^CreER^* both label connective tissue fibroblasts in the developing diaphragm. We have included additional panels to Figure 4—figure supplement 1 of *PDGFRa^CreER/+^; HGF^delta/flox^; Rosa^mTmG/+^*embryos demonstrating the presence of *GFP+* fibroblasts in the muscleless regions. Thus, fibroblasts do expand and appropriately populate the diaphragm even in the absence of muscle.

The phenotype after genetic deletion of *HGF* via *Prx1Cre* or *PDGFRa^CreER^* and after MET-inhibitor treatment is quite similar (see slightly revised model in Figure 8). Genetic *HGF* deletion or MET inhibition early leads to ventral muscleless regions and two bilateral dorsal muscleless regions (Figures 4 C, H and 5 B-D). Genetic *HGF* deletion or MET inhibition later consistently leads to ventral muscleless regions and less commonly to dorsal muscleless regions (Figures 4 G and 5 E). We have provided additional clarification in the text and in model in Figure 8.

3) The authors propose a role of Met in myogenic commitment, based on the co-culture experiments. Myogenic commitment implies a major role in myogenic lineage progression or differentiation. The evidence for that claim is weak and the reduced differentiation might be the consequence of increased cell death, reduced proliferation or other causes that change myoblast density. It is mandatory to address this issue appropriately, by analyzing both, fibroblast and myoblast cell numbers, proliferation, apoptosis, myogenic differentiation (i.e. ratio of MyoD/MyoG-positive cells).

We apologize for the misunderstanding here and have altered the text to indicate that we do not propose a role for Met in myogenic commitment, but rather that Met regulates the number of MyoD+ cells by promoting their survival. Using our co-culture methods, unfortunately we are unable to quantify fibroblast cell numbers, as they are too dense to distinguish using either our software or manual counting.

4) A potentially interesting conclusion form the manuscript is hat hernia does only develop when both, connective tissue and muscle of the diaphragm are affected. Although this explanation is intriguing, the authors have to make sure that body wall muscles are fully developed, to exclude that the abdominal pressure is not reduced in mice conditional Met/HGF mutations. The images in Figures 1 and 2 seem to indicate normal development of body wall muscles but a dedicated statement in this respect would be helpful.

We have added a statement in the Results section indicating the body wall muscles are normal.

5) Treatment of WT embryos from E7.5 will impact on myogenic progenitor delamination, and this is also expected for Pax3CreKIMetnull/fl mutation. The authors have to dissociate the Met-related somite delamination, a migration of the progenitors to the diaphragm and the potential additional role of Met signaling in the colonization and expansion of myogenic progenitor cells in the anlage of the diaphragm.

In Figure 5, we have delivered the Met inhibitor at various stages to determine when Met is required. Delivery at E7.5-E12.5 gives the most pronounced phenotype; muscleless regions in the dorsal and ventral regions of the diaphragm. To exclude that inhibition of Met is affecting the initial delamination of progenitors from the somites and migration to the PPFs, we also delivered the inhibitor at E11.5-E12.5 (Figure 5E). Previously we showed that migration to the PPFs is completed by E11.5 (Sefton et al., 2018). Thus the appearance of dorsal and ventral muscleless regions in these diaphragms demonstrates that Met has an additional later role.

Our original analysis of the mechanistic effects of Met inhibition in vivo (in Figure 7) were based on experiments administering BMS777607 E7.5-E12.5 because this dosage gave the most highly penetrant and consistent phenotype. To exclude possible effects due to early delamination and migration defects, we have repeated these experiments by administering inhibitor at E11.5-E12.5 and analyzing the ventral-most expansion of diaphragm muscle at E15.5 (Figure 7O-V). We find that Pax7/MyoD/Myosin+ cells are substantially reduced in the ventral leading edge of the diaphragm following BMS777607 treatment. This demonstrates that Met is required at later timepoints for the ventral expansion of the diaphragm.

6) Validation for the drug BMS777607 effectiveness in inhibiting MET downstream pathway should be performed and shown.

Due to limited number of cells in our PPF cultures, quantitative validation of pathways is difficult in the embryonic diaphragm. As such, we have validated the effectiveness of BMS777607 in inhibiting phosphorylation of MET, which is essential for downstream pathway signaling, using adult skeletal muscle stem cells, where there are sufficient cells to perform phospho-specific flow cytometry (Figure 6—figure supplement 1). Moreover, the drug BMS777607 has been previously validated in downregulating pAkt, pERK, and pPLC by Western blot (Faham, Zhao and Welm, 2018 doi:10.1038/s41523-018-0091-5). BMS777607 from Selleck Chem has been cited by 47 publications.

7) Comments referring to missing controls, scale bars, statistics need to be addressed.

We have addressed comments about controls, scale bars, and statistics below and thank reviewers for their careful and thoughtful reviews.

Reviewer #1 (Recommendations for the authors):It was previously shown that HGF and Met controls development of the diaphragm muscle. In particular, the signal induces delamination and migration of muscle progenitor cells that colonize the diaphragm. The present manuscript by Sefton and coworkers confirms and extends these observations using (i) conditional mouse lines in which the HGF gene was targeted by Cre/loxP recombination in the pleuroperitoneal folds (Prx1-cre) and at other sites PdgfraCreERT2, and of (ii) Met inhibitors. Overall, the technical quality of the data on diaphragm muscle development is excellent; the conceptual advance over previous work is not exceptional; the evidence for Met/HGF-dependent development of the phrenic nerve is marginal and needs to be strengthened.The data show that fibroblasts provide HGF signals received by Met in muscle progenitor cells that is essential for diaphragm development. The PdgfraCreERT2 line was used to demonstrate that HGF produced by fibroblasts but not by muscle progenitors is essential for diaphragm development. Moreover, development of dorsal and ventral regions of diaphragm muscle requires continuous MET signaling. Thus, HGF is not only required for the delamination of progenitors, but also for proliferation and survival of those muscle progenitors that reached the anlage of the diaphragm.My major concern is the limited data on the HGF-dependent development of the phrenic nerve (defasciculation). While it is well documented that HGF acts as a trophic factor for motor neurons in culture, its role in development of motor neurons was highly debated due to the fact that some changes observed in Met or HGF mutant mice in vivo are also present in other mutants that lack the muscle groups derived from migrating muscle progenitors. Moreover, careful genetic analyses previously demonstrated indirect mechanisms of Met during motor neuron development, i.e. a non-cell-autonomous function of Met during the recruitment of motor neurons to PEA3-positive motor pools (Helmbacher et al., Neuron 2003).Sefton et al., provide an analysis of a single time point, one histological picture (3G, magnified in 3H) that indicate that in Met+/- animals defasciculation of the phrenic nerve does not occur correctly. This is accompanied by a quantification that barely reaches significance (Figure 3K). Data shown in Figure 7 using Met inhibitors show a major change in phrenic nerve branching, which is presumably due to the major change in diaphragm development, as conceded by the authors.Despite this weakness on the experimental side, the role of HGF/Met in phrenic nerve development is strongly emphasized in abstract /intro/discussion (e.g. line 414: However, PPF-derived HGF is crucial for the defasciculation and primary branching of the nerve, independent of muscle). The data need to be strengthened in order to conclude that HGF coordinates both, diaphragm muscle and phrenic development. I expect that the defasciculation of the phrenic nerve is highly dependent on the developmental stage. The authors should provide data that show different stages of phrenic nerve development, i.e. the time course of the of defasciculation in wildtype animals, explain how the staging was done, and compare different stages in the Met mutants and analyze whether the defasciculation is resolved at later stages. Met mutations also affect placental development, resulting in developmental delays that in turn might lead to an apparent small change in the time course of defasciculation. The authors should exclude that indirect effects cause the small change in phrenic nerve morphology, for instance by examining conditional Met mutations that are restricted to motor neurons. Good Cre lines that target motor neurons are available.

In response to comments from the reviewers, we have more thoroughly investigated the role of Met in the development of the phrenic nerve and include two new sets of genetic experiments. In our first submission, we found a decreased number of phrenic nerve branches at E11.5 in *Met ^Δ/ Δ^* and *Met ^Δ/+^* compared with *Met^+/+^* embryos. In the *Met ^Δ/ Δ^* embryos, no muscle is present in the diaphragm. Therefore, the greatly reduced branching in these embryos is likely a secondary effect of the requirement of Met in muscle progenitors for diaphragm muscularization. Of particular interest is the reduced branching in the *Met ^Δ/+^* embryos. Because the diaphragm is muscularized in these embryos, this suggested that Met may be required intrinsically in the phrenic nerve. One reviewer suggested that the reduced branching in the *Met ^Δ/+^* embryos could be due to a developmental delay in the whole embryo. However, we found that *Met ^Δ/ Δ^* and *Met ^Δ/+^* embryos are not overall delayed relative to *Met^+/+^* embryos (as measured by crown rump length or limb length; Figure 3—figure supplement 1). Also, to increase the robustness of these data, we added additional embryos to the analysis. We then extended our analysis of *Met ^Δ/ Δ^*, *Met ^Δ/+^* and *Met^+/+^* embryos to E12.5 (Figure 3—figure supplement 1) to see whether the branching phenotype persisted; surprisingly we found that while the of *Met ^Δ/ Δ^* embryos continue to have very few branches, the number of branches in *Met ^Δ/+^* embryos recovers and matches that of *Met^+/+^* embryos.

To explicitly test whether Met is required within the phrenic nerve, we used *Olig2^Cre/+^*to conditionally delete *Met*. This line was chosen for its early expression in motor neurons (Zawadzka et al., 2010). We examined *Olig2^Cre/+^;Met*
^Δ*/flox*^ embryos compared to *Olig2^Cre/+^; Met^flox/+^* embryos. We chose to include *Olig2^Cre^* in our controls because the *Olig2^Cre^* is a knock-in/knock-out and *Olig2* has important roles in nerve development. However, deletion of *Met* did not affect the number of branches at E11.5 (Figure 3—figure supplement 2) or E12.5 (data not shown). These data suggest that Met does not intrinsically regulate phrenic nerve branching. This suggests that PPF-derived HGF regulates phrenic nerve branching indirectly via muscle. To test if HGF is sufficient to promote early stages of nerve branching in the absence of muscle, we turned to *Pax3^SpD/SpD^* mutants in which a point mutation in Pax3 prevents migration of muscle progenitors into the diaphragm (Figure 3—figure supplement 2). In these embryos, the diaphragm is muscleless, but the PPFs still express HGF. In these diaphragms the number of branches at E11.5 is severely reduced. These data demonstrate that in the absence of muscle the presence of HGF in the PPF fibroblasts is not sufficient to support diaphragm branching.

Altogether our data demonstrate that PPF-derived HGF, via its regulation of muscle, controls the primary branching of phrenic nerve. The *Met ^Δ/+^* data demonstrate that Met controls phrenic nerve branching at E11.5 in a dose-dependent manner, but this effect is lost by E12.5. Although we see no obvious defects in muscle of *Met ^Δ/+^* diaphragms at later stages, the most parsimonious explanation of the reduced phrenic nerve branching at E11.5 is that this is due to fewer muscle progenitors at this time point.

We thank the reviewers for prompting us to look at the role of HGF/Met in the phrenic nerve more closely. Our revised conclusions are presented in the Results and Discussion. We show that PPF-derived HGF is critical for integrating both muscle and phrenic nerve development, but now demonstrate that HGF’s regulation of phrenic nerve branching is via muscle, which is well-known to express multiple trophic factors required by motor neurons.

Reviewer #2 (Recommendations for the authors):Since the authors observed a correlation between loss of should muscle and loss of diaphragm muscularization, which is related to the timing of migration of shoulder muscle and diaphragm muscle progenitors, they claim a “closer relationship” between should muscles and diaphragm, which was further extended in the discussion. I was not convinced by this conclusion. Is there a “closer relationship” between the muscles just because the progenitor cells migrate roughly at the same time? They authors may modify or delete this statement, although I agree that a broader expression of HGF may facilitate enhanced recruitment of muscle progenitor cells, required for formation of the diaphragm.

This statement has been modified (lines 206-211).

“However, based on their similar temporal sensitivity to HGF/MET signaling, the shoulder acromiodeltoid and spinodeltoid progenitors migrate at a similar time as the diaphragm progenitors. Thus continued expression of *HGF* at this later developmental time point may facilitate recruitment of muscle cells necessary for development of diaphragm and shoulder muscles.”

Based on co-cultures of PPFs and myoblasts the authors describe a function of MET in myogenic commitment. In my opinion the evidence for such a function is weak. The authors observed no change in proliferation but reduced motility and a higher rate of apoptosis after pharmacological inhibition of MET. Increased apoptosis and reduced aggregation of myoblasts at distinct locations may easily interfere with myogenic differentiation. A general function of MET in myogenic commitment does not seem very likely, since myoblasts that do not undergo prior long-range migration differentiate normally in the absence of Met.

We apologize for the misunderstanding here and have altered the text to indicate that we do not propose a role for Met in myogenic commitment, but rather that Met regulates the number of MyoD+ cells by promoting their survival.

Pharmacological inhibition of MET increases the rate of apoptosis in numerous cell types, which has been studied extensively in cancer. It seems appropriate to explore the mechanism of increased apoptosis in myoblasts following MET inhibition more closely. Previous reports suggest increased expression of p53, increased sensitivity to Fas-mediated apoptosis or increased autophagy, among others, as potential causes for increased apoptosis after MET inhibition. Which pathway is relevant in myoblasts?

We have performed qPCR on vehicle and BMS777607-treated PPFs for *Fas, Trp53,* and autophagy marker *Map1l3ca*. We found increased expression of *Fas* and *Map1l3ca* in PPFs following treatment with the Met inhibitor, while *Trp53* did not change (Figure 6—figure supplement 1).

Obviously, it would be great to learn more about the mechanisms that control Hgf expression in fibroblasts within and derived from PPTs. Such knowledge may also help to better understand the reasons leading to Congenital Diaphragmatic Hernias (CDH). Unfortunately, the authors did not to go any further in this direction.

Understanding the mechanisms regulating HGF expression in the PPFs is critical for understanding why in development and in evolution muscle is recruited to the nascent diaphragm. This will be explored in future studies.

The authors observed that branching of the phrenic nerve was reduced in heterozygous Met mutants with normal diaphragm musculature, suggesting a direct role of Met in phrenic nerve branching. Timed inactivation of Met specifically in motoneurons would greatly increase the impact of this finding and allow a more specific analysis of the role of MET in phrenic nerve development and branching.

Please see response to Essential Revision Point 1. We have now included a more detailed analysis of the role of Met in phrenic nerve branching, including conditional deletion in motor neurons and analysis of *Pax3^SpD/SpD^* mice.

Surprisingly, the authors did not observe CDH in mutant mice with muscle-less diaphragms, from which they conclude that additional defects in the connective tissue are necessary to allow hernia formation. Although this explanation is intriguing, the authors have to make sure that abdominal pressure is not reduced in mice without muscularized diaphragms, e.g. demonstrate that body wall muscles are fully functional. The images in Figures 1 and 2 seem to indicate normal development of body wall muscles but a dedicated statement in this respect would be helpful.

We have added a dedicated statement to the manuscript indicating that no defects in abdominal wall muscle development were observed.

Scale bars are missing in some panels.

Thank you for this observation and we have checked for scale bars in figure panels. Please note that some figures contain one scale bar for multiple panels.

According to the methods part, Student’s t-test was used for the statistical analysis shown in Figure 3. A pairwise comparison of WT, heterozygous and homozygous Met mutants is not appropriate when all three genotypes are compared with each other. An ANOVA test should be used as in Figure 5F.

We had used a one-way ANOVA for the statistical analysis used in Figure 3, but did not specifically state this in the methods. This has now been added to the methods and Figure legend.

Reviewer #3 (Recommendations for the authors):The present manuscript addresses questions on the role of HGF/MET signaling in diaphragm formation once myogenic progenitors have already migrated to the PPFs. In addition, it identifies the PPFs as the source of HGF. The study is interesting and developed with rigor. However, the role of HGF is not clearly dissociated from the presence/absence of the fibroblasts/connective tissue itself.

We have added to a supplemental figure demonstrating that fibroblasts are indeed present in muscleless regions following loss of HGF (Figure 4—figure supplement 1D-F).

Also, the authors do not link in the results, for example, how the muscleless diaphragms and HGF itself relate to the hernia phenotype mentioned in the abstract.

We now discuss how the partially muscleless diaphragms relate to herniation in the abstract. While loss of muscle is necessary for development of hernias, it is not sufficient; additional defects in the connective tissue are necessary to weaken the diaphragm and lead to CDH.

Figure 11C) Co-staining for Hgf (PPFs) and Met (migrating progenitors) should be provided for a clear visualization of ligand-expressing cells and receptor-expressing cells.

We have included an additional panel in Figure 1—figure supplement 1 showing coexpression of *HGF* and *Met* in E11.5 PPF.

1F) Co-staining for Met and migrating progenitors (PAX3 or LBX1 for example) should be provided for a clear visualization of migrating progenitors versus general Met expression in other cell types.

We have previously shown this in a supplemental figure analyzing the migration pathway of diaphragm muscle progenitors, co-staining MET and GFP abelling in *Pax3^CreKI/+^; Rosa^mTmG/+^* embryos in cross section (Figure S2 in Sefton *et al.,* 2018 doi: 10.1016/j.ydbio.2018.04.010).

1M) The claim that fibroblast-derived HGF is required for diaphragm muscle development cannot be addressed with only this experimental analysis. What is the connective tissue phenotype in PDGFaCreER;HGFnull/fl embryos? Lack of connective tissue could affect muscle migration and development rather than HGF expression on its own. Are fibroblasts still present?

We have demonstrated that fibroblasts are still present in muscleless regions (see point 2 above and Figure 4—figure supplement 1).

1G, I, K, M) Control genotypes should be analyzed and added in the Figure or as Supplementary Data.

We have added a supplemental figure with control embryos stained for myosin (Figure 1—figure supplement 3).

1I) Pax3CreKIMetnull/fl originates a muscleless diaphragm but this could be associated with lack of delamination and migration from the somites rather than a specific MET requirement for diaphragm muscle formation once progenitors have colonized this area as suggested by the authors.

Yes, we agree. It is the primary reason we chose to use the pharmacological inhibitor BMS777607 to control the timing of MET signaling inhibitions.

Figure 3It is not clear whether HGF controls phrenic nerve formation independently of muscle. In the end of this section the authors mention that Met heterozygous embryos have normal muscles referring to Figure 1K (which is not a picture referring to a Met het embryo). The authors should confirm if in Met null embryos there is a direct effect on the nerve bifurcation and brunching or if the lack of muscle is leading to this observation. Is the phenotype dose dependent for muscle formation? This is not properly shown by the authors. Could the authors perform a conditional KO of Met in the nerves?This concern is further supported by the Figure 4 data where phrenic nerve only extends to regions with muscle.

Please see response to Essential Revision Point 1. We have now included a more detailed analysis of the role of Met in phrenic nerve branching, including conditional deletion in motor neurons and analysis of *Pax3^SpD/SpD^* mice.

Figure 4The role of HGF versus the presence of fibroblasts/connective tissue remains vague. Is the lack of ventral muscles associated with lack of migration of progenitors within the forming diaphragm towards an HGF source? Or due to the lack of connective tissue scaffold itself? Why is Prx1Cre used instead of PDGFRaCre (the one used in the actual experiments) to trace PPFs?

We have lineage data demonstrating that *PDGFRa*-derived fibroblasts are present in the muscleless regions. The *Prx1Cre* lineage is also used in supplemental experiments for *HGF* deletion and in previous work from the lab. We believe it is useful to demonstrate the finding that fibroblasts are present with multiple lines of evidence and have left this in the supplemental figure (Figure 4—figure supplement 1).

4E) Please provide a control with a Cre allele to compare putative secondary effects for the presence of the Cre in the cKO embryos.

The panel in 4E has been replaced with a control that has the Cre allele.

Figure 5Validation for the drug BMS777607 effectiveness in inhibiting MET downstream pathway should be performed and shown.

Due to limited cell number, this is difficult to directly measure in embryonic diaphragms either via Western or phospho-flow cytometry. As such, we took a strategy to measure phospho-Met levels in adult skeletal muscle satellite cells. We have demonstrated that phospho-Met levels are downregulated in skeletal muscle satellite cells following treatment with BMS777607 (please see point 6 above; Figure 6—figure supplement 1A).

The phenotype with the inhibition of Met signaling (lack of dorsal muscles in addition to ventral) is distinct to the one observed in Figure 4 (less ventral muscle formation in PDGFaCreER;HGFnull/fl). How do the authors explain the dorsal phenotype when inhibiting Met (since migration should have not been affected at this time-point) and Hgf inhibition in Figure 4 is not leading to this phenotype?

There is quite a bit of variability in the *PDGFRa^CreER/+^; HGF^null/fl^* embryos and we do indeed often find dorsal muscleless regions in *PDGFRa^CreER/+^; HGF^null/fl^* embryos given tamoxifen at E9.5 (see panels 4C and 4H, where myofibers are absent in the dorsal regions of the diaphragm). We suspect that slight differences in the timing of Cre-mediated deletion of *HGF* deletion likely have a substantial impact on the extent of muscle loss and result in the variable muscle phenotypes. We have also clarified the summary of the phenotype of *PDGFRa^CreER/+^; HGF^null/fl^* diaphragms in the model in Figure 8.

Figure 6What is the time point after treatment in B and C? What is the total nuclei number in the cultures (since these are co-cultures)?

The timepoint after treatment is 72 hr. This has been added to the figure legend. With our co-culture method, we have not been able to maintain myogenic progenitors after passaging. As such, these experiments are done on freshly explanted PPFs that are in a pyramid shape on the plate. Because the fibroblasts are tightly packed together in 3-dimensions prior to passaging, we cannot count total nuclei accurately with the automated software or evenly manually. The myogenic progenitors are less dense and can be accurately counted. As such, unfortunately we do not have total numbers of nuclei.

The authors should provide pictures together with nuclear staining for a global view of the cell density in the cultures. Also T0 time point pictures should be added for comparison, in particular in C.

We have added photos of nuclear staining of a 72 hr culture to a new supplemental figure (Figure 6 —figure supplement 2). While we cannot get accurate quantitative data from this, it does give a qualitative picture of overall nuclear density. T0 pictures, using GFP fluorescence of live muscle progenitors have also been added to this supplemental figure.

In these experiments the authors performed co-culture (fibroblasts + myogenic progenitors) derived from PPfs. However, fibroblast phenotype is not addressed. If there is less GFP+ cells is there more fibroblasts in the culture? Could this impact on the phenotype observed and linked by the authors to the Met inhibitor treatment? The only piece of evidence is Gata4 qPCR which is not sufficient to address fibroblast phenotype in the culture.

Although it is possible that fibroblasts are affected by Met inhibition, our in vivo deletion of Met in fibroblasts (Figure 1K) does not suggest that Met inhibition would negatively impact PPF fibroblasts in culture. Because of technical limitations (see above), it is difficult to explicitly quantify the number of fibroblasts unless we were to passage the fibroblasts.

Figure 7Treatment of WT embryos from E7.5 will impact on myogenic progenitor migration. The authors have to dissociate the Met-related somite migration phenotype and the potential additional role of Met signaling in the diaphragm muscle formation.

Our original analysis of the mechanistic effects of Met inhibition in vivo (in Figure 7) were based on experiments administering BMS777607 E7.5-E12.5 because this dosage gave the most highly penetrant and consistent phenotype. To exclude possible effects due to early delamination and migration defects, we have repeated these experiments by administering inhibitor at E11.5-E12.5 and analyzing the ventral-most expansion of diaphragm muscle at E15.5 (Figure 7O-V). We find that Pax7/MyoD/Myosin+ cells are substantially reduced in the ventral leading edge of the diaphragm following BMS777607 treatment. This demonstrates that Met is required at later timepoints for the ventral expansion of the diaphragm.

[Editors’ note: further revisions were suggested prior to acceptance, as described below.]

Reviewer #1 (Recommendations for the authors):The majority of my concerns were appropriately addressed by the authors. There are a few additional points that need to be addressed, and most of these concern the text/wording.Unfortunately, the changes in the text have not been highlighted in red in the uploaded version of the manuscript, as it was mentioned in the rebuttal letter.1. While the quality of the figures is overall good, many of the panels in Figure 2 are very dark. It should be easy to modify this.

Thank you for this observation. We have brightened many of the panels in Figure 2.

2. While the authors show very convincingly that the effects on the phrenic nerve observed are caused indirectly by the loss of muscle, and not directly by the loss of HGF/Met signaling, this is not always made clear in the text. Furthermore, the mechanism that causes branching deficits should be clearly stated in the Abstract.Line 41-42...and indirectly required for phrenic nerve primary branching.Please mention the specific indirect mechanisms, i.e. via the effect on muscle development.

We have explicitly noted the indirect effect is mediated by muscle in the abstract.

Line 81…HGF is also critical for innervation.The reports cited do not distinguish between the effect of the nerve/muscle or report in vitro experiments. The text should take this into account.

A sentence has been added indicating the reports do not distinguish between effect on muscle and nerve.

238: Thus, PPF-derived HGF is necessary for phrenic nerve defasciculation.Additional experiments shown in the next paragraph indicate that it is the absence of the muscle that causes defasciculation. Please rephrase.

This has been rephrased as “Thus, loss of PPF-derived HGF leads to both muscle defects and phrenic nerve defasciculation defects.”

3) Met inhibition using BMS777607BMS777607 acts as an AXL, RON and Met tyrosine kinase inhibitor. This should be mentioned. Are effects of RON/AXL on skeletal muscle development described?

We have added a note that BMS777607 also inhibits AXL and RON in addition to MET. There is limited information about the role of RON/AXL on skeletal muscle development. A recent study found that Axl is a survival and growth receptor in mouse myoblasts during skeletal muscle regeneration (Al-Zaeed et al., 2021; doi: 10.1038/s41419-021-03892-5). A published meeting abstract states that double knock out Gas6-Axl mice have reduced hindlimb skeletal muscle mass (doi: https://doi.org/10.1096/fasebj.2020.34.s1.09757).

4) Antibody specificity: Antibody specificity should be tested by analysis of the phrenic nerve on Met mutants.

The antibody we used is the goat polycolonal (#AF527 R&D Systems) which was generated against a recombinant protein that encompasses Glu25-Asn929. This region includes the extracellular semaphorin domain, furin cleavage site, plexin semaphorin domain, and the four IPT domains. The mutant Met allele we are using is from Huh et al., (PNAS 2004; doi: 10.1073/pnas.0306068101) and deletes exon 16 which contains a critical ATP-binding site in the intracellular tyrosine kinase domain. Western analysis from Webster et al., (Figure 1C; PLoS 2013; doi: https://doi.org/10.1371/journal.pone.0081757) and the Huh et al., 2004 paper (Figure 1 L) show that the MET pTyr 1234/1235 domain and downstream pAKT are inactivated with this mutant allele, respectively. However, the pre-processed MET protein is still present in these mutants (Figure 1K of Huh et al., and Figure 1B of Webster et al.,). The R&D antibody that we and others use recognizes the extracellular domain that is present in the preprocessed protein (see Figure 1B of Webster et al). Thus when we examine *Met^D/D^* mutants using the R&D antibody, we still see labeling (see Author response image 1). Therefore the Huh et al., Met allele that we use is not able to test the specificity of the R&D antibody. However, over 24 references have used this antibody (https://www.rndsystems.com/products/mouse-hgfr-c-met-antibody_af527#product-citations). We have not exhaustively searched these references to determine whether any tested the specificity of the R&D antibody (e.g. that it does not cross-react with the closely related RON protein).

**Author response image 1. sa2fig1:** ­The extracellular region of MET protein (using AF527 R&D antibody) is still detected in *Met^D/D^* mutants. Transverse cross section of E10.5 embryos with dorsal to the top; lateral is on the left side in left panel; lateral is on the right side in right panel. NT, neural tube.

Reviewer #2 (Recommendations for the authors):Sefton et al., have submitted a revised version of a study, in which the role of fibroblast-derived HGF for recruitment and expansion of muscle during morphogenesis of the mammalian diaphragm was investigated. The authors have changed the title to cope with new findings, indicating that reduced primary branching of the phrenic nerve in Met-mutants is not due direct effects of HGF on the nerve but most likely caused by indirect effects resulting from reduced muscle formation in Met-mutants.The authors did an excellent job to deal with the reviewers' criticisms. In particular, by specifically deleting Met in motoneurons and by analysis of splotch mutants they demonstrate that Met does not intrinsically regulate phrenic nerve branching. Analysis of splotch mice, which show normal expression of HGF PPFs but display muscle-less diaphragms, revealed reduced branching of the phrenic nerve, similar to Met-mutants, clearly suggesting a critical role of muscle fibers in the diaphragm for phrenic nerve branching. Furthermore, Sefton et al., now demonstrate that fibroblasts expand and populate the diaphragm in the absence of muscle, which essentially excluded a connective tissue phenotype in conditional Hgf/Met mutants. They also clarified the statement about what I understood was meant to claim a role of Hgf/Met in myogenic commitment. Additional controls (effectiveness of BMS777607 for inhibition of MET, treatment with BMS777607 at additional timepoints) were done as requested, providing additional interesting insights into a role of Met for ventral expansion of the diaphragm at relatively late developmental timepoints.The authors argue that "PPF-derived HGF, via muscle, controls phrenic nerve defasciculation". Well, this is formally correct but exaggerates the role PPF-derived for phrenic nerve defasciculation in my view. Probably, deletion of any gene that prevents diaphragm muscle formation will have similar effects on phrenic nerve defasciculation. I would prefer a more neutral statement: "Muscle formation, requiring PPF-derived HGF, controls phrenic nerve defasciculation", or something similar. The authors may consider modifying the statement.I do not have any further objections.

We appreciate the positive response to our revisions. We have altered the sentence as suggested.